# Hierarchical Skills for Efficient Exploration

**Jonas Gehring**[1,2]   **Gabriel Synnaeve**[1]   **Andreas Krause**[2]   **Nicolas Usunier**[1]

[1]Facebook AI Research   [2]ETH Zürich

`jgehring@fb.com`

## Abstract

In reinforcement learning, pre-trained low-level skills have the potential to greatly facilitate exploration. However, prior knowledge of the downstream task is required to strike the right balance between generality (fine-grained control) and specificity (faster learning) in skill design. In previous work on continuous control, the sensitivity of methods to this trade-off has not been addressed explicitly, as locomotion provides a suitable prior for navigation tasks, which have been of foremost interest. In this work, we analyze this trade-off for low-level policy pre-training with a new benchmark suite of diverse, sparse-reward tasks for bipedal robots. We alleviate the need for prior knowledge by proposing a hierarchical skill learning framework that acquires skills of varying complexity in an unsupervised manner. For utilization on downstream tasks, we present a three-layered hierarchical learning algorithm to automatically trade off between general and specific skills as required by the respective task. In our experiments, we show that our approach performs this trade-off effectively and achieves better results than current state-of-the-art methods for end-to-end hierarchical reinforcement learning and unsupervised skill discovery. Code and videos are available at `https://facebookresearch.github.io/hsd3`.

## 1   Introduction

A promising direction for improving the sample efficiency of reinforcement learning agents in complex environments is to pre-train low-level skills that are then used to structure the exploration in downstream tasks [23, 19, 27, 13, 30]. This has been studied in particular for the control of (simulated) robots, where there is a natural hierarchical decomposition of the downstream tasks into low-level control of the robot's actuators with a skill policy, and a high-level control signal that specifies a direction or target robot configuration with coarser temporal resolution. The large body of work on unsupervised skill or option discovery in hierarchical reinforcement learning (HRL) for continuous control relies, explicitly or implicitly, on prior knowledge that low-level skills should control the center of mass of the robot [26, 17, 13, 43, 6]. This nicely fits a wide range of benchmark tasks that are variants of navigation problems, but the benefit of such hierarchical setups outside this problem class is unclear.

The prior knowledge embedded in a pre-trained skill defines a specific trade-off between sample efficiency and generality. Skills that severely constrain the high-level action space to elicit specific behavior (e.g., translation of the center of mass) are likely to provide the largest gains in sample efficiency, but are unlikely to be useful on a diverse set of downstream tasks. Conversely, low-level skills that expose many degrees of freedom are more widely applicable but less useful for guiding exploration. There is, thus, no single universally superior pre-trained skill. Depending on the downstream task, different skills might also be useful to efficiently explore in different parts of the environment.

In this paper, we aim to acquire skills that are useful for a variety of tasks while still providing strong exploration benefits. We propose to pre-train a *hierarchy of skills* of increasing complexity which can subsequently be composed with a high-level policy. In the context of simulated robots, each skill consists of controlling a part of the robot configuration over a short time horizon, such

35th Conference on Neural Information Processing Systems (NeurIPS 2021).

as the position of the left foot of a humanoid, or the orientation of its torso. Skills of increasing complexity are constructed by jointly controlling larger portions of the configuration. These skills are modelled with a shared policy and pre-trained in an environment without rewards and containing the robot only. Subsequently, skills are used in downstream tasks within a three-level hierarchical policy: the highest level selects the skill (which specifies a *goal space*), the second level the target configuration within that skill (the *goal*), and the pre-trained skill performs the low-level control to reach the goal. Compared to standard approaches involving a single static pre-trained skill [13, 43], our approach offers increased flexibility for structuring exploration and offloads the issue of selecting prior knowledge from pre-training to downstream task learning. As a result, our skills can be acquired once per robot and applied to many different tasks.

We perform an experimental analysis of our hierarchical pre-training on a new set of challenging sparse-reward tasks with simulated bipedal robots. Our experiments show that each task is most efficiently explored by a distinct set of low-level skills, confirming that even on natural tasks, where locomotion is of primal importance, there is no overall single best pre-trained skill. We further show that dynamic selection of goal spaces with a three-level hierarchy performs equally or better than a generic skill on all tasks, and can further improve over the best single skills per task.

The main contributions of our work are summarized as follows:

- We propose a novel unsupervised pre-training approach that produces a hierarchy of skills based on control of variable feature sets.

- We demonstrate how to automatically select between different skills of varying complexity with a three-level hierarchical policy that selects skills, goals, and native actions.

- We introduce a benchmark suite of sparse-reward tasks that allows for consistent and thorough evaluation of motor skills and HRL methods beyond traditional navigation settings.

- We study the implications of prior knowledge in skills experimentally and showcase the efficacy of our hierarchical skill framework on the proposed benchmark tasks, achieving superior results compared to existing skill discovery and HRL approaches [13, 33, 53].

## 2   Related Work

The success of macro-operators and abstraction in classic planning systems [14, 40] has inspired a large body of works on hierarchical approaches to reinforcement learning [8, 46, 3, 50, 33, 39]. While the decomposition of control across multiple levels of abstraction provides intuitive benefits such as easier individual learning problems and long-term decision making, recent work found that a primary benefit of HRL, in particular for modern, neural-network based learning systems, stems from improved exploration capabilities [21, 35]. From a design perspective, HRL allows for separate acquisition of low-level policies (options; skills), which can dramatically accelerate learning on downstream tasks. A variety of works propose the discovery of such low-level primitives from random walks [26, 52], mutual information objectives [17, 13, 43, 6], datasets of agent or expert traces [37, 2, 25], motion capture data [36, 31, 42], or from dedicated pre-training tasks [15, 27].

In order to live up to their full potential, low-level skills must be useful across a large variety of downstream tasks. In practice, however, a trade-off between generality (wide applicability) and specificity (benefits for specific tasks) arises. A large portion of prior work on option discovery and HRL resolved this trade-off, explicitly or implicitly, in favor of specificity. This can be exemplified by the choice of test environments, which traditionally revolve around navigation in grid-world mazes [46, 8, 10]. In recent work on simulated robotics, navigation problems that are similar in spirit remain the benchmarks of choice [12, 33, 13, 27, 2]. In these settings, directed locomotion, i.e., translation of the robot's center of mass, is the main required motor skill, and the resulting algorithms require a corresponding prior. This prior is made explicit with skills partitioning the state space according the agent's position [13, 43], or is implicitly realized by high contribution of position features to reward signals [33, 34, 2] (Appendix G). Similarly, works that target non-navigation environments acquire skills with tailored pre-training tasks [38] or in-domain motion capture data [36, 31, 32]. In contrast, our work is concerned with learning low-level skills without extra supervision from traces or pre-training task design, and which do not prescribe a fixed trade-off towards a particular type of behavior.

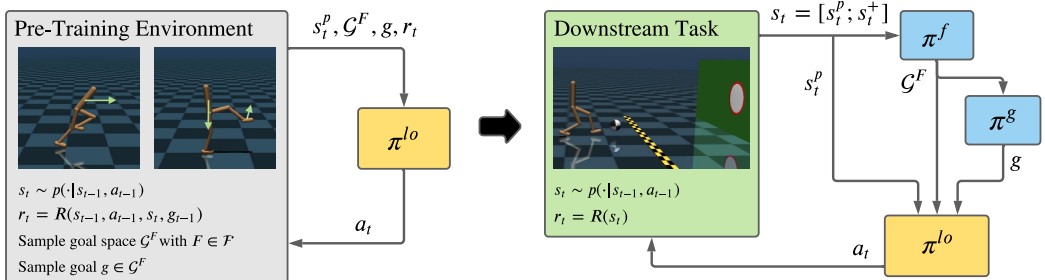

Figure 1: Illustration of our proposed hierarchical skill learning framework. **Left** Low-level policies are learned in an empty pre-training environment, with the objective to reach random configurations (goal $g$) of a sampled skill (goal space $\mathcal{G}^F$ defined over a feature set $F$). Examples of goal space features are translation along the X-axis or the position of a limb. **Right** Learning on downstream tasks with a three-level hierarchical policy to select a goal space, a goal and finally a native action $a_t$ with the pre-trained low-level policy. The low-level policy acts on proprioceptive states $s^p$, while high-level policies $\pi^f$ and $\pi^g$ leverage extra task-specific information via $s^+$.

## 3 Hierarchical Skill Learning

### 3.1 Overview

In this work, we propose an approach where we first acquire low-level policies that are able to carry out a useful set of skills in an unsupervised manner, i.e., without reward signals from the main tasks of interest. We subsequently employ these skills in a hierarchical reinforcement learning setting (Figure 1). The pre-training environment consists solely of the robot. In downstream tasks, additional objects may be present besides the robot; these may be fixed (e.g., obstacles) or only indirectly controllable (such as a ball).

Crucially, the usefulness of a skill policy stems from providing additional structure for effective exploration. In order to provide benefits across a wide range of tasks, skills need to support both fast, directed exploration (e.g., locomotion) as well as precise movements (e.g., lift the left foot while bending down). We propose to fulfill these requirements with short-horizon, goal-directed low-level policies that are trained to achieve target configurations of robot-level state features such as the position or orientation of its torso or relative limb positions. We denote this feature space with $\mathcal{S}^g$. This allows the definition of a hierarchy of skills by controlling single features and their combinations, resulting in varying amounts of control exposed to high-level policies. Each skill is trained to reach goals in a goal space $\mathcal{G}^F$ defined over a set of features $F$ of $\mathcal{S}^g$, yielding policies of the form

$$\left\{\pi^{\mathrm{lo}}_F : \mathcal{S}^{\mathrm{p}} \times \mathcal{G}^F \rightarrow A\right\}_{F \in \mathcal{F}}$$

with proprioceptive observations $\mathcal{S}^{\mathrm{p}}$ (Figure 1, left). Goal-directed skill policies are trained without task-specific rewards, relying solely on state features related to the specific robot and the prescribed hierarchy of goal spaces.

On downstream tasks, high-level policies operate with a combinatorial action space as the skill policy $\pi^{\mathrm{lo}}$ is conditioned on both a feature set $F$ and a concrete goal $g \in \mathcal{G}^F$. State spaces are enriched with task-specific features $\mathcal{S}^+$, s.t. $\mathcal{S} = \mathcal{S}^{\mathrm{p}} \cup \mathcal{S}^+$, that contain information regarding additional objects in the downstream task. This enriched state space is only available to the high-level policy. We model the high-level policy in a hierarchical fashion with two policies

$$\pi^{\mathrm{f}} : S \rightarrow \mathcal{F}, \quad \pi^{\mathrm{g}} : S \times \mathcal{F} \rightarrow \mathcal{G}^F,$$

that prescribe a goal space and a goal, respectively (Figure 1, right). With a task- and state-dependent policy $\pi^{\mathrm{f}}$, it is possible to not only select the required set of skills for a given environment, but also to switch between them within an episode. In this three-level hierarchy, the higher-level policy $\pi^{\mathrm{f}}$ explores the hierarchy of goal spaces and dynamically trades off between generality and specificity. Temporal abstraction is obtained by selecting new high-level actions at regular intervals but with a reduced frequency compared to low-level actions. We term the resulting learning algorithm *HSD-3*, emphasizing the three different levels of control (goal spaces, goals and native actions) obtained after our hierarchical skill discovery phase.

## 3.2 Unsupervised Pre-Training

During pre-training, skill policies act in an MDP $\mathcal{M}$ with proprioceptive states $s^p$, native actions $a$, transition probabilities and initial states $s_0^p \in \mathcal{S}_0^p$ drawn from a distribution $P_0$ [45, for example], but without an extrinsic reward function. The skill policies are trained to reach goals in a goal space defined over $S^g$ consisting of robot-level sensor readings, which may include non-proprioceptive information.

Critically, each skill $\pi_F^{lo}$ aims to achieve goals defined over a variable-sized subset of features $F$ of $S^g$. To avoid learning a combinatorial number of independent policies, we follow the principle of universal value function approximators [41] and share the parameters of each skill. We augment the policy's input with goal and goal space information, and learn a single set of parameters $\theta$, leading to $\pi_\theta^{lo} : \mathcal{S}^p \times \mathcal{F} \times \mathcal{G} \to \mathcal{A}$, where $\mathcal{F} := \{F : F \subseteq \mathcal{S}^g\}$ and $\mathcal{G} := \{\mathcal{G}^F : F \subseteq \mathcal{S}^g\}$. For ease of modeling, $F$ is provided to $\pi_\theta^{lo}$ as a bag-of-words input where a coordinate is set to 1 if its respective feature is included in $F$ and to 0 otherwise. Also, the low-level policy receives goals relative to the current values of the $s^g$. Thus, at each time step, the input goal to the policy is updated to reflect the progress towards the goal. As a learning signal, we provide a distance-based reward as $R(s^g, a, s^{g'}, F, g) := ||\omega^F(s^g) - g||_2 - ||\omega^F(s^{g'}) - g||_2$, where $\omega^F : \mathcal{S}^g \to \mathcal{G}^F$ is a fixed transformation that selects the subset $F$ of goal space features and applies a suitable normalization. The selection of features for $\mathcal{S}^g$ and transformations $\omega^F$ represents the prior knowledge that we utilize for unsupervised pre-training. We discuss examples for bipedal robots in our experiments and in Appendix B.

During training, features $F$ and goals $g$ are sampled anew for each episode in the environment. Episodes consist of several dozen steps only, reflecting the fact that we are interested in short-horizon skills. In-between episodes, we reset the simulation to states drawn from $\mathcal{S}_0^p$ only sporadically to encourage the resulting policies to be applicable in a wide range of states, facilitating their sequencing under variable temporal abstractions. Likewise, rewards are propagated across episodes as long as the simulation is not reset. The low-level policy parameters $\theta$ can be optimized with any reinforcement learning algorithm; here, we opt for Soft Actor-Critic (SAC), a state-of-the-art method tailored to continuous action spaces [18]. Pseudo-code for the pre-training algorithm and further implementation details are provided in Appendix C.

## 3.3 Hierarchical Control

After a low-level policy $\pi^{lo}$ has been obtained via unsupervised pre-training, we employ it to construct a full hierarchical policy $\pi(a|s) = \pi^f(F|s)\pi^g(g|s, F)\pi^{lo}(a|s^p, F, g)$. The hierarchical policy is applied to downstream tasks, where we are supplied with a reward signal and state observations $\mathcal{S} = \mathcal{S}^p \cup \mathcal{S}^+$. While different frequencies may be considered for acting with $\pi^f$ and $\pi^g$, in this work both high-level actions are selected in lock-step. For jointly learning $\pi^f$ and $\pi^g$, we extend the SAC algorithm to incorporate the factored action space $\mathcal{F} \times \mathcal{G}$. In particular, it is necessary to extend SAC to deal with a joint discrete and continuous action space, which we describe below (see Appendix D for further details and pseudo-code for high-level policy training).

We formulate our extension with a shared critic $Q(s, F, g)$, incorporating both factors of the action space. $Q(s, F, g)$ is trained to minimize the soft Bellman residual [18, Eq. 5], with the soft value function $V(s)$ including entropy terms for both policies. Since $\pi^f$ outputs a distribution over discrete actions (to select from available goal spaces), we compute its entropy in closed form instead of using the log-likelihood of the sampled action as for $\pi^g$. We employ separate temperatures $\alpha$ and $\beta^F$ for the goal space policy as well as for each corresponding goal policy; $\beta^F$ are normalized to account for varying action dimensions across goal spaces.

$$V(s) = \sum_{F \in \mathcal{F}} \pi^f(F|s) \mathop{\mathbb{E}}_{g \sim \pi^g} \left[ Q(s, F, g) - \frac{\beta^F}{|F|} \log \pi^g(g|s, F) \right] + \alpha \left( \mathcal{H}(\pi^f(\cdot|s)) - \log |\mathcal{F}| \right).$$

Compared to the usual SAC objective for continuous actions, we subtract $\log |\mathcal{F}|$ from the entropy to ensure a negative contribution to the reward. This matches the sign of the entropy penalty from $\pi^g$, which is negative in standard hyper-parameter settings[1] [18]. Otherwise, the entropy terms for discrete and continuous actions may cancel each other in Q-function targets computed from $V(s)$.

---

[1]The entropy for continuous actions sampled from a Gaussian distribution can be negative since it is based on probability densities.

The policy loss [18, Eq. 7] is likewise modified to include both entropy terms. Parameterizing the policies with $\phi$ and $\psi$ and the Q-function with $\rho$, we obtain

$$J_{\pi^{f,g}}(\phi,\psi) = \mathop{\mathbb{E}}_{s \sim B} \left[ \sum_{F \in \mathcal{F}} \pi^{\mathrm{f}}_{\phi}(F|s) \mathop{\mathbb{E}}_{g \sim \pi^{\mathrm{g}}_{\psi}} \left[ \frac{\beta^F}{|F|} \log \pi^{\mathrm{g}}_{\psi}(g|F,s) - Q_{\rho}(s,F,g) \right] - \alpha\,\mathcal{H}(\pi^{\mathrm{f}}_{\phi}(\cdot|s)) \right],$$

where the replay buffer is denoted with $B$. The scalar-valued entropy temperatures are updated automatically during learning to maintain target entropies $\overline{\mathcal{H}}^f$ and $\overline{\mathcal{H}}^g$ for both $\pi^{\mathrm{f}}$ and $\pi^{\mathrm{g}}$. The following losses are minimized at each training step:

$$J(\alpha) = \mathop{\mathbb{E}}_{s \sim B} \left[ \alpha \left( \mathcal{H}(\pi^{\mathrm{f}}(\cdot|s)) - \overline{\mathcal{H}}^f \right) \right]$$

$$J(\beta) = \mathop{\mathbb{E}}_{s \sim B} \left[ -\sum_{F \in \mathcal{F}} \beta^F \pi^{\mathrm{f}}(F|s) \mathop{\mathbb{E}}_{g \sim \pi^{\mathrm{g}}} \left[ \frac{1}{|F|} \log \pi^{\mathrm{g}}(g|s,F) + \overline{\mathcal{H}}^g \right] \right].$$

We implement temporal abstraction by selecting high-level actions with a reduced frequency. This incurs a reduction in available training data since, when taking a high-level action every $c$ steps in an $N$-step episode, we obtain only $N/c$ high-level transitions. To leverage all available transitions gathered during training, we adopt the step-conditioned critic proposed by Whitney et al. [52] in our SAC formulation. The Q-function receives an additional input $0 \leq i \leq c$ marking the number of steps from the last high-level action, and is trained to minimize a modified soft Bellman residual:

$$J_Q(\rho) = \mathop{\mathbb{E}}_{\substack{F_t, g_t, i, \\ s_{t,\ldots,t+c-i}, \\ a_{t,\ldots,t+c-i} \sim B}} \left[ \frac{1}{2} \left( Q_{\rho}(s_t, F_t, g_t, i) - \left( \sum_{j=0}^{c-i-1} \left( \gamma^j r(s_{t+j}, a_{t+j}) \right) + \gamma^{c-i} V(s_{t+c-i}) \right) \right)^2 \right]$$

As in [52], $Q(s,F,g,0)$ is used when computing $V(s)$ and the policy loss $J_{\pi^{f,g}}(\phi,\psi)$.

The factorized high-level action space of HSD-3 has previously been studied as parameterized action spaces [28, 51], albeit with a small number of discrete actions and not in the context of hierarchical RL. A possible application of Soft Actor-Critic is described in Delalleau et al. [9]; our approach differs in that we (a) compute the soft value function as a weighted sum over all discrete actions, and (b) opt for two separately parameterized actors and a shared critic. These design choices proved to be more effective in initial experiments.

# 4  Benchmark Environments

We propose a benchmark suite for comprehensive evaluation of pre-trained motor skills, tailored to bipedal robots (Figure 2) and implemented for the MuJoCo physics simulator [49, 48]. We selected tasks that require diverse abilities such as jumping (Hurdles), torso control (Limbo), fine-grained foot control (Stairs, GoalWall) and body balance (PoleBalance). Besides PoleBalance, all tasks also require locomotion. The tasks are designed to have a sparse reward that is provided once an obstacle or step has been passed or a goal has been shot, emphasizing the need for good exploration. As an exception, PoleBalance provides a constant reward during the entire episode, which ends when the pole placed on the robot falls over. PoleBalance requires precise motor control, and is, in combination with its reward structure, chiefly outside of the realm of benchmarks that have been traditionally used to evaluate HRL techniques. Obstacle position and step lengths are subject to random perturbations, sampled anew for each episode. We tailor several environment parameters to the concrete robot being controlled, e.g., the positions of Limbo bars. For the effectively two-dimensional Walker robot, we limit the movement of objects (ball, pole) to translation along X and Z and rotation around the Y-axis. In all environments, we label simulation states that would cause the robot to fall over as invalid; these will terminate the episode with a reward of -1. Appendix A contains full specifications for all tasks.

Variants of some of our benchmark tasks have been proposed in prior works: Hurdles [20, 24, 13, 38], Stairs [29, 16], Gaps [20, 48], and PoleBalance is inspired by the classical CartPole problem [4]. Our tasks are different from earlier published counterparts in parameterization and in the sparsity of rewards they provide. All environments are provided via a standard Gym interface [5] with accompanying open-source code, enabling easy use and re-use.

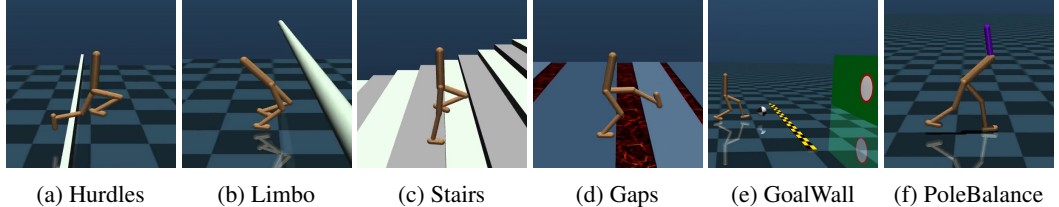

| (a) Hurdles | (b) Limbo | (c) Stairs | (d) Gaps | (e) GoalWall | (f) PoleBalance |
|---|---|---|---|---|---|

Figure 2: Benchmark environments for evaluating motor skills for bipedal robots, pictured with the Walker robot. Hurdle and Limbo bar heights and spacing, as well as stair lengths, are sampled randomly from fixed distributions. We also include a combination of Hurdles and Limbo (HurdesLimbo), in which both obstacle types alternate.

## 5 Experimental Results

For our experiments, we first pre-train skill policies as described in Section 3.2 in an empty environment. The feature set to describe target configurations consists of the robot's translation along the X-axis, its position on the Z-axis, torso rotation around the Y-axis and positions of each foot, relative to the body (Appendix B). With these five features, we obtain a set of $2^5 - 1 = 31$ skills with a shared set of policy parameters. We then train separate high-level policies for each benchmark environment, directing the same set of pre-trained skills. High-level actions are taken every 5 environment steps; in the PoleBalance environment, which requires short reaction times, all three policies operate at the same frequency. All neural network policies employ skip connections as proposed for SAC by Sinha et al. [44]. We show results over 9 seeds for each run; in experiments involving pre-training, we combine 3 seeds for pre-training with 3 seeds for high-level policy training. Pre-training takes approximately 3 days on 2 GPUs (V100) and fine-tuning (downstream task training) takes 2 days to reach 5M samples on 1 GPU. Further details regarding the training setup, hyper-parameters for skill and high-level policy training, as well as for baselines, can be found in Appendix E.

### 5.1 Trade-offs for Low-Level Skills

In a first experiment with the Walker robot, we aim to investigate the trade-off between specificity and generality of low-level skills. For each environment, we train high-level policies directing a *fixed* skill within its respective goal space, which for example consists of torso-related features only, or of feet positions combined with forward translation. At the end of training, we measure average returns obtained for all 31 skills in a deterministic evaluation setting, and plot quartiles over different seeds for the best skills in Figure 3. The performance for different fixed skills varies significantly across environments, and *no single skill is able to obtain the best result (shaded) across all environments*. While controlling X, Y and Z features only produces the good hierarchical policies for most tasks, poor results are obtained in Gaps and GoalWall. Controlling feet positions (LF and RF) is crucial for GoalWall, although close-to-optimal performance is only achieved in few runs across the respective

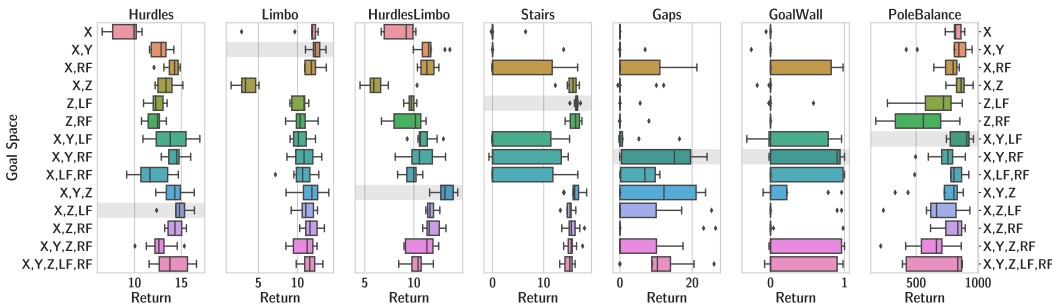

Figure 3: Returns achieved after 5M samples on the benchmark tasks with the Walker robot with fixed low-level policy goal spaces (quartiles). Each row corresponds to a set of features for the respective goal space. Best skills (marked) differ significantly across tasks.

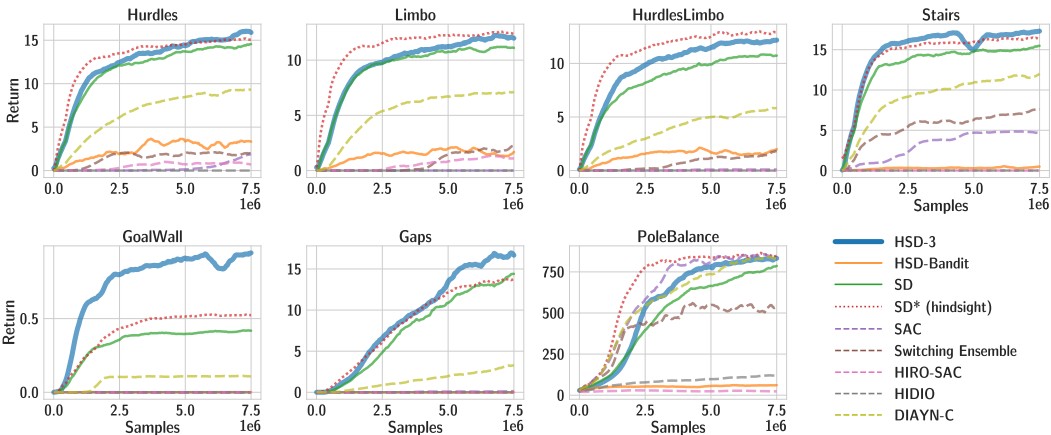

Figure 4: Learning curves on benchmark environments with the Walker robot. For clarity, we plot mean performance over 9 seeds, averaged over 0.5M samples. Full learning curves including error bands are provided in Appendix F.1.

skills. The most general skill, high-level control of *all* considered features (bottom row), also allows for learning progress on all tasks; however, it is generally outperformed by other, more specific skills.

## 5.2 Skill Selection with HSD-3

We compare HSD-3 against a number of base- and toplines to examine the effectiveness of our proposed framework for skill learning and hierarchical control. We include state-of-the-art non-hierarchical RL methods (SAC [18]), end-to-end hierarchical RL algorithms (HIRO [33], HIDIO [53]), and evaluate DIAYN as an alternative unsupervised pre-training algorithm [13]. For HIRO, we use SAC for learning both high- and low-level policies (HIRO-SAC). We facilitate the comparison to DIAYN with a continuous-valued skill variable obtained from an embedding [1] (DIAYN-C). For both HIRO and DIAYN-C, we provide the same goal space as in our pre-training stage, consisting of the full feature set, to ensure a fair comparison. We further compare to the Switching Ensemble proposed in [35], which does not learn a high-level policy but has been shown to improve exploration. Finally, we examine how our pre-trained skills perform if we use a single goal space only (SD). Without prior knowledge about the task at hand, this corresponds to the full goal space $F = \{0, 1, ..., \dim(\mathcal{S}^g)\}$, settling for maximum generality. As a topline, we select the *best per-task* goal spaces from Section 5.1, denoted with SD*, which required exhaustive training and evaluation with all available goal spaces.

In Figure 4, we plot learning curves over 5M training interactions for HSD-3 as well as base- and toplines, and list final performances in Table 1. On all tasks, HSD-3 outperforms or matches (Limbo) the learning speed of a conventional hierarchical policy with a skill defined over the full goal space (SD). On GoalWall and Stairs, the *three-level policy significantly exceeds the mean performance of single goal space base- and toplines*. For Gaps, the variances of the best results in Table 1 (HSD-3,

| Method | Hurdles | Limbo | HurdlesLimbo | Stairs | Gaps | GoalWall | PoleBalance |
|---|---|---|---|---|---|---|---|
| SAC | 2.2±6.2 | -0.1±0.2 | -0.0±0.4 | 5.0±4.8 | 0.1±0.5 | -0.2±0.3 | **866.8**±104.5 |
| Switching Ensemble | 0.7±3.0 | 1.9±4.3 | 1.6±3.6 | 7.4±3.8 | -0.2±0.3 | -0.2±0.3 | 569.4±230.1 |
| HIRO-SAC | 0.4±1.6 | 0.9±2.2 | -0.0±0.1 | -0.0±0.0 | -0.1±0.2 | -0.0±0.0 | 23.8±12.4 |
| HIDIO | -0.1±0.1 | -0.1±0.1 | -0.2±0.1 | -0.2±0.3 | -0.2±0.3 | -0.3±0.3 | 117.6±33.8 |
| DIAYN-C | 9.5±2.5 | 7.5±1.3 | 5.7±1.7 | 12.5±2.9 | 3.4±4.1 | 0.1±0.3 | **839.9**±58.4 |
| SD | **15.0**±1.4 | 11.0±1.0 | 10.8±1.4 | 15.2±0.9 | **14.5**±9.3 | 0.4±0.5 | 789.5±79.1 |
| HSD-Bandit | 2.6±1.3 | 2.4±1.7 | 2.0±1.5 | 0.5±0.8 | -0.1±0.2 | -0.1±0.2 | 61.0±15.8 |
| HSD-3 | **15.3**±2.0 | **11.7**±0.9 | **11.8**±1.3 | **17.2**±0.7 | **15.1**±8.9 | **0.9**±0.1 | **876.0**±36.9 |
| SD* | **15.0**±1.0 | **12.2**±1.0 | **12.7**±1.3 | 16.2±1.3 | **14.0**±11.3 | 0.5±0.5 | **868.2**±91.1 |

Table 1: Final performance after 7.5M samples across benchmark tasks with the Walker robot. Mean and standard deviation reported for average returns over 9 seeds.

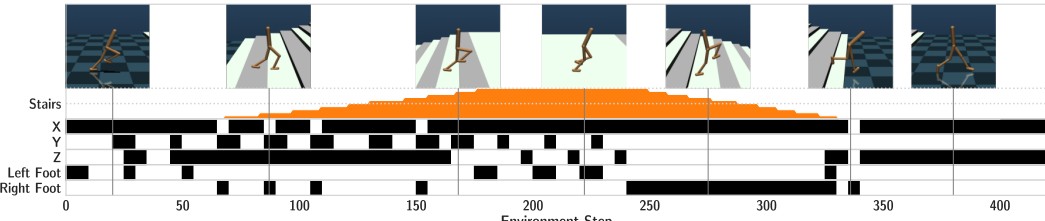

Figure 5: Skills selected by HSD-3 during an episode on the Stairs task. Different skills are utilized for different sections of the environment: walking upstairs is achieved by controlling the torso position (X,Z) in a regular pattern, with goals for the rotation (Y) and the right foot being set occasionally. On the top section, a different pattern is selected for quickly running forwards. When going downstairs, the right foot of the robot is controlled explicitly to maintain balance.

SD, SD*) are high, hinting at especially challenging exploration problems. A closer look at per-seed performance (Appendix F.1) reveals learning failure for 2 out of 9 runs for these methods. For GoalWall, 4 and 5 runs for SD* and SD, respectively, achieve zero return. Per-task goal space selection with a bandit algorithm (HSD-Bandit) fails on most environments, and achieves low returns on Hurdles, Limbo and HurdlesLimbo.

These results demonstrate that, without additional task-specific knowledge, our hierarchical skill framework is able to *automatically and efficiently trade off between skills of varying complexity at training time*. In Figure 5, we visualize the goal spaces that are selected across the course of an episode in the Stairs environment. During different stages of the environment (walking upstairs, downstairs, and on a planar surface), HSD-3 selects appropriate skills of varying complexity[2].

The performance of the remaining baselines underscores the challenges posed by our benchmark tasks. Standard SAC works well in PoleBalance, which has a dense reward structure compared to the remaining environments. On most other tasks, learning progress is slow or non-existent, e.g., on Hurdles, only 2 out of 9 runs achieve a positive return at the end of training. DIAYN-C exhibits subpar performance across compared our hierarchical pre-training scheme, with the exception of PoleBalance where learning is fast and returns are higher than for SD. The end-to-end HRL methods HIRO-SAC and HIDIO are unable to make meaningful progress on any task, which highlights the utility of skill pre-training in the absence of task-specific priors. The Switching Ensemble is the best baseline without pre-training (except for the PoleBalance task), but a clear majority of individual runs do not obtain positive returns (Appendix F.1). On the Stairs task, both SAC and the Switching Ensemble manage to climb the flight of stairs in several runs, but fail to discover the second, downwards flight at the end of the platform.

We refer to the supplementary material for further ablation and analysis with the Walker robot. In Appendix F.2, we demonstrate the efficacy of our proposed hierarchical pre-training method, compared to pre-training a single skill policy on the full goal space exclusively. In Appendix F.3, we analyze the exploration the behavior of various methods from Table 1. We find that, in general, hierarchical methods visit more states compared to SAC. HIRO-SAC and DIAYN-C visit a higher number of states in the Hurdles, Limbo, HurdlesLimbo and Stairs environments but fail to leverage this experience for achieving higher returns.

### 5.3 Evaluation on a Humanoid Robot

To evaluate the scalability of HSD-3 to more complex robots, we perform an additional set of experiments with the challenging 21-joint Humanoid robot from Tassa et al. [48]. For skill pre-training, we select a goal space approximately matching the Walker setup, consisting of the translation along the X-axis of simulation, Z position of the torso, its rotation around the Y axis and three-dimensional feet positions relative to the robot's hip location.

In Figure 6, we plot average learning progress across 100M environment steps, comparing HSD-3 to SD (the full goal space), SD* (the best goal space), and a conventional, non-hierarchical SAC setup. The results align with the observations for the Walker robot in Section 5.2 in that HSD-3

---

[2]Videos are available at `https://facebookresearch.github.io/hsd3`.

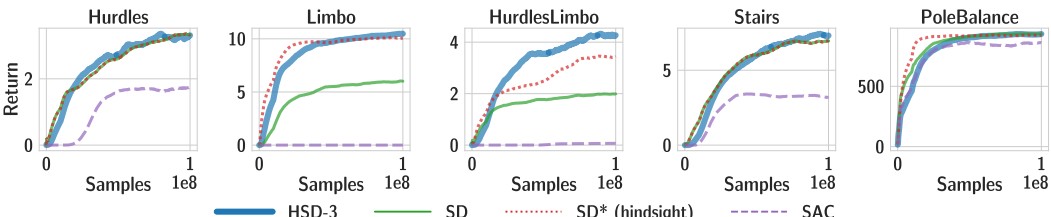

Figure 6: Results on four of the benchmark tasks with a 21-joint Humanoid robot. We show mean performance over 9 seeds, clipped to 0 and averaged over 2.5M samples. Appendix F.5 contains learning curves for all seeds and error bands.

matches or surpasses the performance of the full goal space, and can further outperform the best goal space selected in hindsight. Both HSD-3 and SD make good learning progress on three out of five tasks used for this evaluation (Hurdles, Stairs, PoleBalance); on Hurdles and Stairs, SD is in fact the best single goal space. On HurdlesLimbo, which requires both jumping and crouching, HSD-3 outperforms both SD and SD*. On the Limbo task, HSD-3 is able to learn faster than SD and achieves a higher return by utilizing the most effective single skills on this task, which control X translation and leaning forward or backward. This behavior was not discovered by the SD baseline, which exercises control over all goal space features. In this preliminary study, none of the hierarchical policies was able to make significant progress on the GoalWall and Gaps tasks, which were already shown to be difficult with the Walker robot in Section 5.2.

## 6 Conclusion

Our work is among the first to highlight the specific trade-offs when embedding prior knowledge in pre-trained skills, and to demonstrate their consequences experimentally in simulated robotic control settings. We describe a hierarchical skill learning framework that, compared to existing approaches, allows high-level policies to perform the trade-off between directed exploration and fine-grained control automatically during training on downstream tasks. Our experiments with a bipedal Walker robot demonstrate HSD-3's efficacy on a variety of sparse-reward tasks in which previous approaches struggle. We further apply our framework to a challenging Humanoid robot, where it learns effective policies on the majority of tasks.

With this work, we provide a new set of benchmark tasks that require diverse motor skills and pose interesting challenges for exploration. We release the tasks in a dedicated distribution, with the aim of spurring further research on motor skills in continuous control settings, and to ultimately broaden the utility of hierarchical approaches to control to tasks beyond navigation.

**Limitations:** Our investigations with the Humanoid robot have been performed in limited goal spaces and hence with significant priors. We believe that further work is required to successfully acquire a large, exhaustive set of low-level skills in unsupervised environments that work on more and more complex morphologies.

**Acknowledgements:** We thank Alessandro Lazaric for insightful discussions, and Franziska Meier, Ludovic Denoyer, and Kevin Lu for helpful feedback on early versions of this paper.

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
