# A  Environment Details

For all benchmark tasks, we place the robot at a designated starting position and configuration, and perturb its joint positions with noise sampled uniformly from $[-0.1, 0.1]$ and its joint velocities with noise sampled from $0.1 \cdot \mathcal{N}(0,1)$. These perturbations are also applied in standard MuJoCo benchmark tasks [5]. Each environment provides three different observations: proprioceptive robot states $\mathcal{S}^{\mathrm{p}}$, extra task-specific observations $\mathcal{S}^{+}$, and measurements for goal states $\mathcal{S}^{\mathrm{g}}$. Goal state features are solely used for convenience when providing relative goal inputs to low-level policies (Algorithm 2), and can also be derived from proprioceptive observations and the robot's absolute position. Below, we list detailed environment configurations and reward functions, as well as robot-specific modifications (if applicable). In all cases, we define invalid states on a per-robot basis (Appendix B) that lead to a premature end of the episode with a reward of $-1$. Unless otherwise noted, episodes consist of 1000 steps of agent interaction.

**Hurdles** Hurdles take the form of simple boxes, placed in intervals $\sim \mathcal{U}(3,6)$ meters, and with heights $\sim \mathcal{U}(0.1, 0.3)$ meters. Task-specific observations are the distance to the next hurdle and its height. For every hurdle that the robot's torso passes, the agent receives a reward of 1.

**Limbo** Cylindrical bars with a diameter of 0.2m are placed in intervals $\sim \mathcal{U}(3,6)$ meters. Their heights are draw from $\mathcal{U}(1.2, 1.5)$ for the Walker robot, and from $\mathcal{U}(0.9, 1.2)$ for the Humanoid. The agent observes the distance to the next limbo bar, as well as its height above the ground. A reward of 1 is obtained if the agent's torso position moves past a bar.

**HurdlesLimbo** This is a combination of the Hurdles and Limbo environments, where hurdles and limbo bars alternate, starting with a hurdle. The reward is defined as above, and the observation consists of the type of obstacle (0 for a hurdle, 1 for a limbo bar) as well as its relative distance and height.

**Stairs** This task consists of 10 stairs (upwards), a platform, and another 10 stairs (downwards). Stairs have a height of 0.2m, and their length is sampled uniformly within [0.5m,1.0m]. The agent observes the distance of the robot's torso to the next two steps, and a reward of 1 is provided whenever it moves past a step.

**Gaps** We sample gaps from $\mathcal{U}(0.2, 0.7)$ meters and platforms from $\mathcal{U}(0.5, 2.5)$ meters ($\mathcal{U}(1.0, 2.5)$ meters for the Humanoid). Gaps and platforms are placed in alternation, starting at a distance of 4 meters from the robot's initial position. Gaps are placed 5cm below the platforms, and the episode ends with a reward of $-1$ if the robot touches a gap. The agent observes the relative distances to the start of the next gap and next platform. A reward of 1 is achieved if the robot's torso moves past the start of a platform for the first time.

**GoalWall** A ball is placed 2.5 meters in front of the robot (1m for the Humanoid). A wall is placed 4 meters from the ball, and contains a circular target area with an 0.8m diameter, placed 1m above ground. For the effectively two-dimensional Walker robot, ball movements are restricted to rotation around the Y-axis and translation along X and Z. At the start of the episode, the ball's position is perturbed with additive noise drawn from $\mathcal{N}(0, 0.01)$, and we add noise from $\mathcal{N}(0, 0.1)$ to its rotation. The agent observes the position of the ball (X and Y are relative to the robot's torso, Z is absolute) and its velocity. Episodes end after 250 steps or if the ball's X position reaches the position of the wall, with a reward of 1 if the ball is inside the target area and 0 otherwise.

**PoleBalance** A cylindrical pole with a mass of 0.5kg is attached to the robot's topmost body part. The pole has a length of 0.5m, and the position and velocity of the rotational joint that connects it to the robot is perturbed with additive noise from $\mathcal{N}(0, 0.01)$. For the Walker robot, the pole can rotate around the Y-axis only; for the Humanoid robot, rotations around all axes are permitted. The position and velocity of the joint connecting the pole to the robot are provided as observations. The reward is 1 unless the pole is about to fall over, which terminates the episode. A falling pole is defined by the distance along the Z axis between its lower and upper parts falling below 80% of its length.

In Table 2 we provide overview renderings of the Hurdles, Limbo, HurdlesLimbo, Stairs and Gaps environments to illustrate the positioning of objects along the course. In these tasks, for which locomotion along the X-axis is essential, non-hierarchical baselines agents perform well with a simple shaped reward for movement along the X-axis. For environments with randomly generated obstacles such as ours, the effectiveness of a simple "forward" reward has previously been observed by Heess et al. [20].

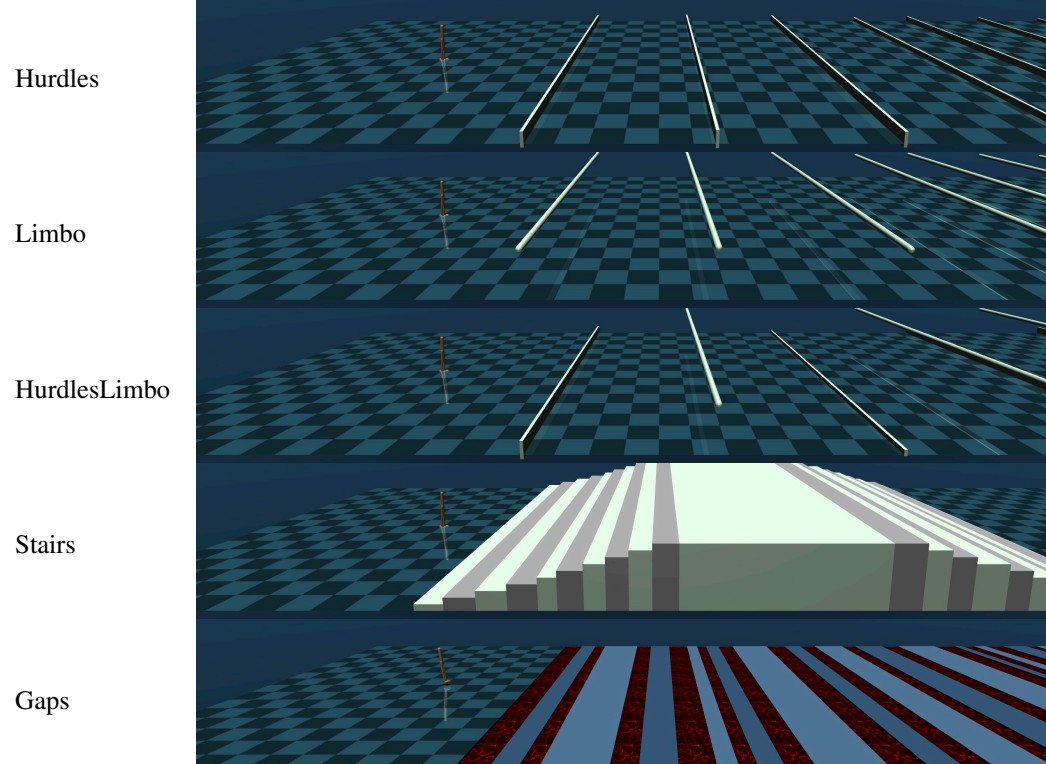

| | Hurdles |
| | Limbo |
| | HurdlesLimbo |
| | Stairs |
| | Gaps |

Table 2: Rendering of 5 of our 7 benchmark tasks. Positions of additional objects (hurdles, limbo bars, gaps, stairs) are subject to random perturbation for each episode. Courses for Hurdles, Limbo, HurdlesLimbo and Gaps continue further.

# B   Robots and Goal Spaces

## B.1   Walker

This two-dimensional, bipedal walker has been released as part of the dm_control suite [48]. We use an observation space similar to the MuJoCo tasks in Gym [5], where we featurize the positions of all modeled joints, their velocities, and contact forces for all body parts clipped to $[-1, 1]$. We manually construct five primitive goal spaces that can be freely combined to obtain a total of 31 candidate goal spaces:

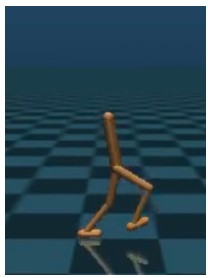

| Feature | | Range (min,max) | Direct Obs. |
|---|---|---|---|
| Torso X position | | $-3, 3$ | no |
| Torso Y rotation | | $-1.3, 1.3$ | yes |
| Torso Z position | | $0.95, 1.5$ | yes |
| Left foot, relative to torso | X pos. | $-0.72, 0.99$ | no |
| | Z pos | $-1.3, 0$ | no |
| Right foot, relative to torso | X pos. | $-0.72, 0.99$ | no |
| | Z pos. | $-1.3, 0$ | no |

Directly observable features are included in proprioceptive observations; non-observable ones can either be derived (i.e., the foot position can be determine via joint positions) or correspond to global observations (X position). All position features are measured in meters while rotations are expressed in radians. Ranges were obtained via rollouts from standard SAC policies trained with shaping rewards.

For the Walker robot, we define invalid states as the torso's Z position falling below 0.9 meters, or its Y rotation having a value outside of $[-1.4, 1.4]$. These limits are intended to prevent the robot from falling over.

## B.2 Humanoid

We adopt the simplified Humanoid robot from Tassa et al. [48], which consists of 21 joints. We use a modified observation space with the same type of state features as the Walker robot (B.1). Following recent work on skill learning for Humanoid robots [36, 31], actions consist of target joint positions, mapped to $[-1, 1]$. We found this to produce better results compared to specifying raw torques.

For goal space featurization, closely follow our Walker setup and provide torso X and Z positions, Y rotation and relative foot positions. Feature values for rotations around Y and Z were obtained with a twist-and-swing decomposition of the torso's global orientation matrix [11]. State features for learning agents use a standard quaternion representation, however.

| Feature | | Range (min,max) | Direct Obs. |
|---|---|---|---|
| Torso X position | | $-3, 3$ | no |
| Torso Y rotation | | $-1.57, 1.57$ | no |
| Torso Z position | | $0.95, 1.5$ | yes |
| Torso Z rotation | | $-1.57, 1.57$ | no |
| | X pos. | $-1, 1$ | no |
| Left foot, relative to hip | Y pos. | $-1, 1$ | no |
| | Z pos. | $-1, 0.2$ | no |
| | X pos. | $-1, 1$ | no |
| Right foot, relative to hip | Y pos. | $-1, 1$ | no |
| | Z pos. | $-1, 0.2$ | no |

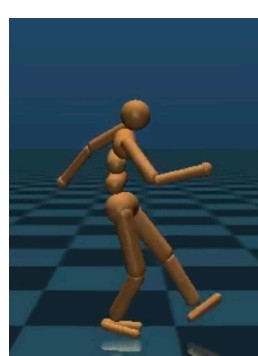

By coupling every feature combination with the Z rotation feature, we again obtain $2^5 - 1 = 31$ possible goal spaces as for the Walker robot.

Similar to the Walker robot, we define valid states as the torso's position being at least 0.9 meters above ground. Body rotations are not restricted.

## B.3 Goal Space Construction

Our main considerations for formalizing goal spaces are the elimination of bias due to different feature magnitudes and a convenient notation for considering subsets of features from the goal feature space $\mathcal{S}^g$. Starting from a set of $n := \dim(S^g)$ features with ranges $(l^i, h^i) : 1 \le i \le n$, we construct a goal space transformation matrix $\Psi$ along with offsets $b$:

$$\Psi := 2I_n \begin{bmatrix} (h^1 - l^1)^{-1} \\ (h^2 - l^2)^{-1} \\ \vdots \\ (h^n - l^n)^{-1} \end{bmatrix}, \; b := -2 \begin{bmatrix} l^1(h^1 - l^1)^{-1} \\ l^2(h^2 - l^2)^{-1} \\ \vdots \\ l^n(h^n - l^n)^{-1} \end{bmatrix} - 1$$

A single goal space over features $F = \{i, j, k, \ldots\}$ is defined as the image of an affine function $\omega^F : \mathcal{S}^g \to \mathcal{G}^F$,

$$\omega^F(s) := s \begin{bmatrix} \Psi_{i,*} \\ \Psi_{j,*} \\ \Psi_{k,*} \\ \vdots \end{bmatrix}^T + \begin{bmatrix} b_i \\ b_j \\ b_k \\ \vdots \end{bmatrix}$$

$\omega^F$ can be understood as simple abstraction that selects a subset $F$ of goal space features and applies a suitable normalization, mapping $[l^i, h^i]$ to $[-1, 1]$ for $i \in F$. A desirable effect of this normalization is that distance-based rewards that are commonly used to train goal-based policies will also be normalized, which facilitates optimization when sharing the parameters of policies across several goal spaces. Additionally, the resulting high-level action space for subgoals can be conveniently defined over $[-1, 1]^F$.

## C  Unsupervised Pre-Training Details

Our unsupervised pre-training algorithm is provided in Algorithm 1. We assume that the pre-training environment provides access to both proprioceptive states (the input of the skill policy) and goal state features as defined in Appendix B. During training, goal spaces and goals are randomly selected for each episode The low-level skill policy $\pi^{\text{lo}}$ and the corresponding Q-functions are trained with the standard Soft Actor-Critic update [18, Alg. 1], with representations of both goal features $F$ and goal $g$ considered a part of the state observations.

The STEP_ENV function in Algorithm 1 samples a transition from the pre-training environment, computes the reward as described in  Section 3.2 and determines the values of done and reset. done is set to true if one of the following conditions hold:

- The current goal is reached (i.e., the L2 distance to it is smaller than $\epsilon_g$).
- A fixed number of steps (horizon $h$) has passed since the previous goal was sampled.
- A random number sampled from $[0, 1)$ is smaller than the resample probability $\epsilon_r$.

reset is set to true if any of these conditions hold:

- The state that was reached is considered invalid (A). In this case, the reward is set to $-1$.
- A fixed number of goals (reset interval $n_r$) have been sampled without a simulation reset.

---

**Algorithm 1** Unsupervised Pre-Training of Hierarchical Skills, see Figure 1 for context

---

**Require:** Goal spaces defined via feature sets $\mathcal{F}$ and transformations $\omega^F$
1: Pre-training environment $E$ with $\mathcal{S} = \mathcal{S}^{\text{p}} \cup \mathcal{S}^{\text{g}}$
2: Initialize policy $\mu_\theta : \mathcal{S}^{\text{P}} \times \mathcal{F} \times \mathcal{G} \to \mathcal{A}$
3: Initialize Q-function $Q_{\phi_i} : \mathcal{S}^{\text{P}} \times \mathcal{F} \times \mathcal{G} \times \mathcal{A} \to \mathbb{R}$ for $i \in \{1, 2\}$
4: Replay buffer $B \leftarrow \text{CIRCULARBUFFER}()$
5: reset $\leftarrow$ true
6: **for** $i \leftarrow 1, N$ **do**
7:     **if** reset **then**
8:         $s \sim S_0$                                              ▷ Reset simulation
9:     **end if**
10:    $F \sim \mathcal{F}, g \sim [-1, 1]^F$                   ▷ Sample new goal space and goal
11:    reset $\leftarrow$ false, done $\leftarrow$ false
12:    **while not** (done **or** reset) **do**
13:        $a \sim \mu_\theta(s^p, F, (\omega^F)^{-1}(g) - s^g)$
14:        $s', r, \text{done}, \text{reset} \leftarrow \text{STEP\_ENV}(s, a, F, g)$
15:        $B . \text{append}(s^p, F, (\omega^F)^{-1}(g) - s^g, a, r, s')$
16:        $s \leftarrow s'$
17:        **if** $i \% f_u == 0$ **then**
18:           **for** each gradient step **do**
19:              $\phi, \theta \leftarrow \text{SAC\_UPDATE}(\phi, \theta, B)$       ▷ Perform standard SAC update
20:           **end for**
21:        **end if**
22:    **end while**
23: **end for**
24: **Output:** $\theta, \phi_1, \phi_2$

---

## D  Hierarchical Control Details

### D.1  Soft-Actor Critic for HSD-3

Below, we provide explicit derivations for extending Soft Actor-Critic to a factorized action space $\mathcal{F} \times \mathcal{G}$ for $\pi^{\text{hi}}$, consisting of discrete actions $F \in \mathcal{F}$ (goal space feature sets) and continuous actions $g \in \mathcal{G}^F$ (goals). Our extension is performed according to the following desiderata: (1) We utilize a shared critic $Q(s, F, g)$ and two policies $\pi^{\text{f}} : \mathcal{S} \to \mathcal{F}, \pi^{\text{g}} : \mathcal{S} \times \mathcal{F} \to \mathcal{G}^F$; (2) we compute $\pi^{\text{f}}$ for all

| Parameter | Value (Walker) | Value (Humanoid) |
|---|---|---|
| Optimizer | Adam [22] | Adam |
| Learning rate $\lambda_Q$ | 0.001 | 0.001 |
| Learning rate $\lambda_\pi$ | 0.001 | 0.001 |
| Learning rate $\lambda_\alpha$ | 0.001 | 0.001 |
| Target entropy $\overline{\mathcal{H}}$ | $-\dim(\mathcal{A}) = -6$ | $-\dim(\mathcal{A}) = -21$ |
| Initial temperature $\alpha$ | 0.1 | 0.1 |
| Target smoothing coefficient $\tau$ | 0.005 | 0.005 |
| Control cost $\zeta$ | 0.01 | 0 |
| Horizon $h$ | 72 | 72 |
| Discount factor $\gamma$ | $1 - 1/h$ | $1 - 1/h$ |
| Goal threshold $\epsilon_g$ | 0.1 | 0.1 |
| Resample probability $\epsilon_r$ | 0 | 0.01 |
| Reset interval $n_r$ | 100 | 100 |
| Replay buffer size | $3 \cdot 10^6$ | $3 \cdot 10^6$ |
| Parallel environments | 20 | 40 |
| Environment steps per iteration | 1000 | 5000 |
| Gradient steps per iteration | 50 | 50 |
| Mini-batch size | 256 | 1024 |
| Warmup steps | $10^4$ | $10^4$ |
| Total iterations | $10^4$ | $3.7 \cdot 10^4$ |

Table 3: Hyper-parameters for unsupervised pre-training.

discrete actions, which provides us with better value estimates a richer policy learning signal; (3) separate, automatically adjusted temperature coefficients are used for the discrete-action policy ($\alpha$) and continuous-action policy ($\beta$). As the action space of $\pi^\mathrm{g}$ is conditioned on $F$, we use separate coefficients $\beta^F$; (4) we further normalize the entropy contributions from the different action spaces $\mathcal{G}^F$ by their dimensionality $|F|$, computing action log-probabilities as $|F|^{-1} \log \pi^\mathrm{g}(g|s, F)$.

The soft value function $V(s)$[18, Eq. 3] for acting with both policies $\pi^\mathrm{f}$ and $\pi^\mathrm{g}$ is given as follows:

$$V(s) = \mathop{\mathbb{E}}_{\substack{F \sim \pi^\mathrm{f}(\cdot|s), \\ g \sim \pi^\mathrm{g}(\cdot|s,F)}} \left[ Q(s, F, g) - \alpha \log \pi^\mathrm{f}(F|s) - \frac{\beta^F}{|F|} \pi^\mathrm{g}(g|s, F) \right]$$

Computing the expectation over discrete actions $F$ explicitly yields

$$V(s) = \sum_{F \in \mathcal{F}} \pi^\mathrm{f}(F|s) \left( \mathop{\mathbb{E}}_{g \sim \pi^\mathrm{g}} \left[ Q(s, F, g) - \frac{\beta^F}{|F|} \log \pi^\mathrm{g}(g|s, F) \right] - \alpha \log \pi^\mathrm{f}(F|s) \right)$$

$$= \sum_{F \in \mathcal{F}} \pi^\mathrm{f}(F|s) \mathop{\mathbb{E}}_{g \sim \pi^\mathrm{g}} \left[ Q(s, F, g) - \frac{\beta^F}{|F|} \log \pi^\mathrm{g}(g|s, F) \right] + \alpha \, \mathcal{H}(\pi^\mathrm{f}(\cdot|s))$$

We arrive at the formulation in Section 3.3 by subtracting the entropy of the uniform discrete-action policy, $\log |\mathcal{F}|$, from $\mathcal{H}(\pi^\mathrm{f}(\cdot|s))$ to ensure negative signage. We proceed in an analogous fashion for policy [18, Eq. 7] and temperature [18, Eq. 18] losses.

The resulting high-level policy training algorithm is listed in Algorithm 2. For brevity, loss and value function definitions above use a single variable for states which indicates both proprioceptive observations $s^p$ and task-specific features $s^+$.

# E  Training Details

**Neural Network Architecture** For all experiments, we use neural networks with 4 hidden layers, skip connections and ReLU activations [44]. Neural networks that operate on multiple inputs (such as the skill policy, or the Q-function in HSD-3) are provided with a concatenation of all inputs. For the Q-function in HSD-3 and HSD-Bandit, goal inputs occupy separate channels for each goal space. The goal policy in HSD-3 $\pi^\mathrm{g}$ is modelled as a multi-head network to speed up loss computations.

**Algorithm 2** HSD-3 High-level Policy Training, see Figure 1 for context

---

**Require:** Low-level policy $\pi_\theta^{\text{lo}}$
**Require:** Goal spaces defined via feature sets $\mathcal{F}$ and transformations $\omega^F$
**Require:** High-level action interval $c \in \mathbb{N}^+$
1: Initialize policies $\pi_\phi^{\text{f}}, \pi_\psi^{\text{g}}$, Q-functions $Q_{\rho_i}$ $i \in \{1, 2\}$, target networks $\overline{\rho_1} \leftarrow \rho_1, \overline{\rho_2} \leftarrow \rho_2$
2: Replay buffer $B \leftarrow \text{CIRCULARBUFFER}()$
3: $s \sim S_0,\ t \leftarrow 0$
4: **for** each iteration **do**
5:     **for** each environment step **do**
6:         **if** $t\%c == 0$ **then**                                                                ▷ Take high-level action
7:             $F \sim \pi_\phi^{\text{f}}([s^p; s^+])$
8:             $g \sim \pi_\psi^{\text{g}}([s^p; s^+], F)$
9:             $g' \leftarrow (\omega^F)^{-1}(g) - s^g$                     ▷ Backproject goal to $\mathcal{S}^{\text{g}}$ and obtain delta
10:         **end if**
11:         $a \leftarrow \mathbb{E}_a\left[\pi_\theta^{\text{lo}}(a|s^p, F, g')\right]$                ▷ Act with deterministic low-level policy
12:         $s' \sim p_E(s'|s, a)$                              ▷ Sample transition from environment
13:         $B\,.\,\text{append}(s, F, g, a, t, r(s, a), s')$
14:         $g' \leftarrow s^g - s'^g + g'$                             ▷ Update goal
15:         $s \leftarrow s'$
16:         $t \leftarrow t + 1$
17:         **if** end of episode **then**
18:             $s \sim S_0,\ t \leftarrow 0$
19:         **end if**
20:     **end for**
21:     **for** each gradient step **do**
22:         Sample mini-batch from $B$, compute losses            ▷ See Section 3.3
23:         Update network parameters $\phi, \psi, \rho_1, \rho_2$
24:         Update temperatures $\alpha, \beta$
25:         Update target network weights $\overline{\rho}_1, \overline{\rho}_2$
26:     **end for**
27: **end for**
28: **Output:** $\psi, \phi, \rho_1, \rho_2$

---

Rather than receiving an input for the selected current goal space, the output of the respective head is selected.

**Hyper-Parameters** are listed inTable 3 for pre-training and Table 5 for HSD-3 high-level policy training. Downstream task training runs with fixed, single goal spaces (SD, SD*) use identical hyper-parameters, but do not require $\lambda_f$, $\overline{\mathcal{H}}^f$, and $\alpha$. Baseline runs use a learning rate of 0.003 for neural networks and an initial temperature of 0.1 [44]. For HSD-3 and SD, we searched for learning rates in $\{0.0001, 0.0003\}$ and initial temperature values in $\{0.1, 1\}$.

**Evaluation Protocol** In regular intervals (Table 5), we perform 50 trials with a deterministic policy and measure the average return that was achieved across trials. Since initial states and environment configurations are randomly sampled, we use the same set of 50 environment seeds for all evaluations to ensure comparability.

## E.1  Baselines

### E.1.1  HIRO-SAC

We implement a end-to-end HRL method similar to HIRO [33], but with Soft Actor-Critic as the underlying learning algorithm rather than TD3. High-level policy actions are expressed within the goal space that we use for pre-training skill policies (B). The neural network policies we use for SAC predict both mean and variances of a Gaussian distribution, and hence we perform goal relabelling by maximizing the log-probabilities of low-level actions directly instead of approximating them via squared action differences [33, Eq. 5]. We observed worse performance by including DynE critic

updates [52]. We therefore report results with an update schedule similar to [33], where high-level policies are being updated less frequently than low-level ones (depending on the high-level action frequency).

### E.1.2  DIAYN-C

In a setup similar to Achiam et al. [1], we learn a continuous representation of DIAYN's discrete skill variable. We input the one-hot encoded skill index to a linear layer with 7 outputs, which corresponds to $\dim(\mathcal{S}^g)$ for the walker. Its output is subject to a hyperbolic tangent activation so that the final low-level policy skill input is bounded in $[-1, 1]$. We operate DIAYN's discriminator on our pre-defined goal spaces (B) and hence provide the same prior knowledge as in our methods. After pre-training with DIAYN, we train a high-level policy as we do for SD baselines, providing its actions directly to the pre-trained policy. We ran experiments with 256 or 1024 hidden units for the skill policy, and with 5,10,20,50 and 100 discrete skils for pre-training. We found that best overall performance was achieved with 256 hidden units for the low-level policy and 10 different skills.

### E.1.3  Switching Ensemble

As proposed by Nachum et al. [35], this baseline consists of a small number of standard policies that gather shared experience and are randomly selected for short time-spans during rollouts. In our version, we use SAC as the underlying learning algorithm, in the same configuration as for the SAC baseline. We use an ensemble of 5 policies, and perform switches with the same frequency that high-level actions are taken at for the other hierarchical methods. For evaluations, we act with a single policy throughout the entire episode.

### E.1.4  HIDIO

We use the official implementation from Github[3]. We found it non-trivial to present the low-level policy discriminator with observations in our standard goal space. Hence, in contrast to HIDIO-SAC and DIAYN-C, the HIDIO baseline discriminator operates on the full state space. In accordance with the other hierarchical methods considered, the steps per option were set to 1 on PoleBalance and 5 otherwise, and the discount factor was set to 0.99. Likewise, we use 256 hidden units for the low-level policy's neural network layers. We performed a hyper-parameter sweep on the Hurdles task, similar to the one performed in the original paper [53] (Table 4).

| Parameter | Value |
|---|---|
| Discriminator input | state, action, state_action, **state_difference** |
| Latent option vector dimension (D) | **8**, 12 |
| Rollout length | 25, 50, **100** |
| Replay buffer length | 50000, **200000** |

Table 4: Hyper-parameters considered for HIDIO, with best ones emphasized.

---

[3]https://github.com/jesbu1/hidio/tree/245d758

| Parameter | Value (Walker) | Value (Humanoid) |
|---|---|---|
| Optimizer | Adam [22] | Adam |
| Learning rate $\lambda_Q$ | 0.001 | 0.001 |
| Learning rate $\lambda_f$ | 0.003 | 0.001 |
| Learning rate $\lambda_g$ | 0.003 | 0.003 |
| Learning rate $\lambda_\alpha$ | 0.001 | 0.001 |
| Learning rate $\lambda_\beta$ | 0.001 | 0.001 |
| Target entropy $\overline{\mathcal{H}}^f$ | $0.5 \log |\mathcal{F}|$ | $0.5 \log |\mathcal{F}|$ |
| Target entropy $\overline{\mathcal{H}}^g$ | $-1$ | $-1$ |
| Initial temperatures $\alpha, \beta^F$ | 1 | 1 |
| Target smoothing coefficient $\tau$ | 0.005 | 0.005 |
| Discount factor $\gamma$ | 0.99 | 0.99 |
| Replay buffer size | $10^6$ | $2 \cdot 10^6$ |
| Parallel environments | 1 | 5 |
| Environment steps per iteration | 50 | 500 |
| Gradient steps per iteration | 50 | 50 |
| Mini-batch size | 256 | 512 |
| Warmup steps | $10^3$ | $10^4$ |
| Evaluation interval (iterations) | 1000 | 400 |

Table 5: Hyper-parameters for high-level policy training with HSD-3.

## F  Extended Results

Below, we provide full learning curves for the results presented in Section 5 and additional experiments.

### F.1  Walker Learning Curves

Learning curves for baselines, HSD-3, HSD-Bandit and SD are provided in Table 6. In addition to the discussion in the experimental section, these plots emphasize that exploration is challenging in all environments apart from PoleBalance. For SAC, SE and HIRO-SAC in particular, positive returns are obtained with few seeds only (gray lines).

### F.2  Ablation: Multi-Task Pre-Training

We perform an ablation study on the impact of our multi-task pre-training algorithm used to obtain a hierarchy of skills (C). We train SD high-level policies, i.e., with a goal space consisting of all considered features, with a skill policy that was trained to reach goals defined in this goal space only. This is in contrast to the pre-trained models used in the other experiments throughout the paper, which are trained to achieve goals within a hierarchy of goal spaces. Networks used to train the single goal-space skill policy consist of 256 units per hidden layer, while skill policies shared among multiple goal spaces use 1024 units (increasing the number of parameters for the single skill policy resulted in worse performance). The results in Figure 7 show that shared policies obtained with multi-task pre-training yield higher returns in most benchmark environments. For GoalWall, the usage of a single-skill policy prevented learning progress altogether. These findings indicate the effectiveness of the multi-trask pre-training stage: it not only produces a compact representation of various skills, making downstream usage more practical, but also results in improved policies for individual goal spaces.

### F.3  Analysis: Exploration Behavior

For gaining additional insight into how HSD-3 impacts exploration behavior, we analyze state visitation counts over the course of training. Due to the continuous state spaces of our environments, we estimate the number of unique states visited over the course of training with SimHash, a hashing method originally proposed for computing exploration bonuses [7, 47]. We hash the full observation, which can include other, randomly placed objects (such as Hurdles or a ball). We compare the amount of unique hashed states for HSD-3 and selected baselines with the Walker robot in Figure 8.

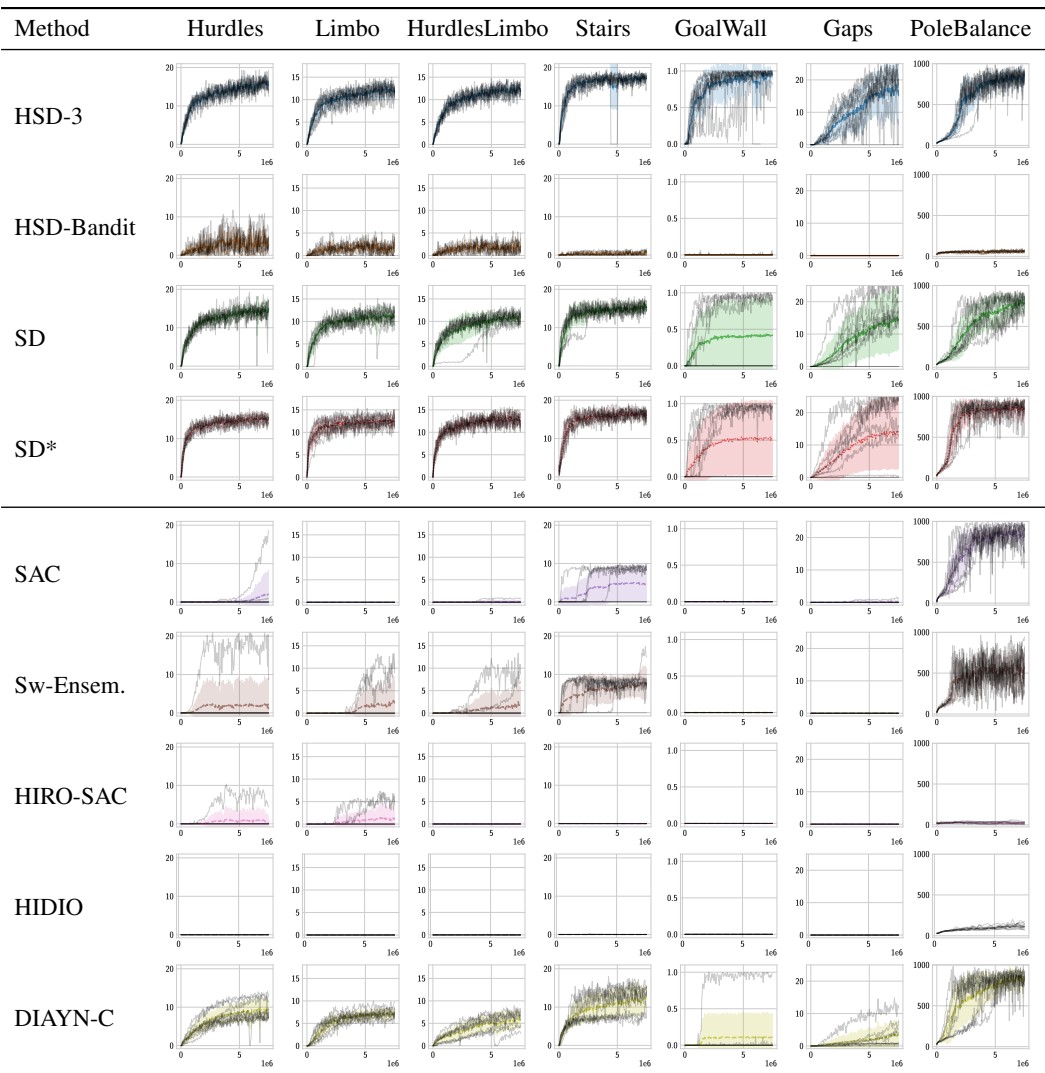

Table 6: Full learning curves for results reported in Table 1 (Walker). We show mean returns achieved (Y axis) after interacting with an environment for a given number of steps (X axis). Shaded areas mark standard deviations computed over 9 seeds for each run.

Generally, hierarchical methods encounter more states during training compared to SAC, even if this does not necessarily translate to a higher return (cf. Figure 4). For HSD-3, after an initial phase of fast learning in Hurdles, Limbo, HurdlesLimbo and Stairs, the performance in terms of return converges, which is reflected in a decrease of new states being visited.

## F.4  Analysis: Pre-Training Performance

In Figure 9, we plot performance during unsupervised pre-training. We train a set goal-reaching policies for different feature combinations, modeled with a single neural network. We found that in our pre-training setup, the training reward alone does not adequately capture the fraction of goals that can be reached reliably. As the number of feature subsets increases, dedicated evaluation runs needlessly prolong the wall-clock training time. As an alternative measure of performance, we train an additional Q-function on a 0/1 reward (1 if a goal was reached) and query it periodically with initial states from the replay buffer and randomly sampled goals. The resulting metric reflects controllability, i.e., the probability with which a randomly sampled goal can be reached.

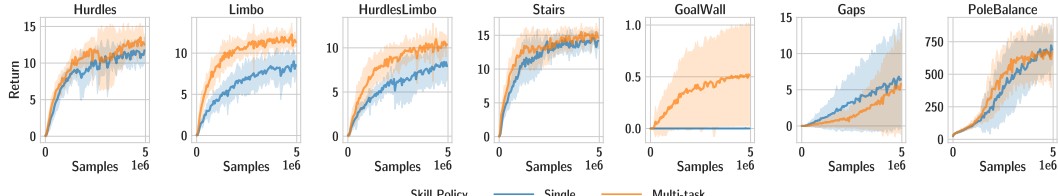

Figure 7: Returns achieved after 5M samples with the SD baseline and the Walker robot, using low-level policies that were trained on a single goal space or with our proposed multi-task pre-training scheme.

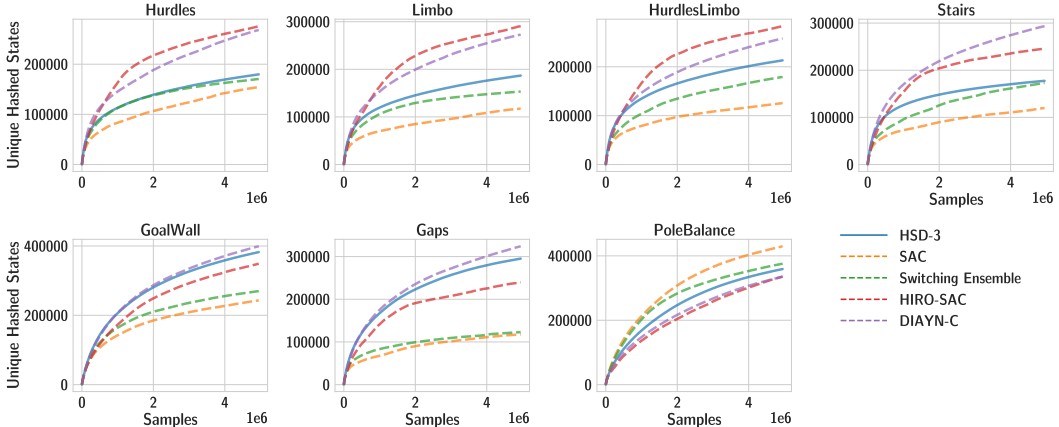

Figure 8: Number of unique hashed states encountered over the course of training (Walker). Mean over 9 seeds per task and method.

Figure 9 shows that, with an increasing number of features, goals become harder to reach. While it is unsurprisingly harder to achieve goals in many dimensions, another effect is that we do not account for fundamentally unreachable states in our definition of the goal space $S^g$. For example, the reachable states for the feet (LF, RF), which are two-dimensional features (X and Z position) roughly describe a half moon while we sample goals from a hypercube. This effect is multiplied when feature sets are combined.

### F.5   Humanoid Learning Curves

In Figure 10, we plot performance for high-level policy training with fixed, individual goal spaces on the Humanoid robot. Similar to the results for the Walker robot (Figure 3), the best goal space features differ significantly across tasks.

Learning curves with the Humanoid robot are presented in Table 7. The non-hierarchical SAC baseline achieves positive returns in the Stairs environment only. The two successful runs manage to climb up the initial flight of stairs but fall over at the top, thus not making it through the entire course. The mediocre median performance of HSD-3 on Stairs is attributed to one of the three seeds failing to learn; the other two are on par with the SD baseline. Notably, HSD-3 outperforms the best single goal space on the Limbo and HurdlesLimbo tasks.

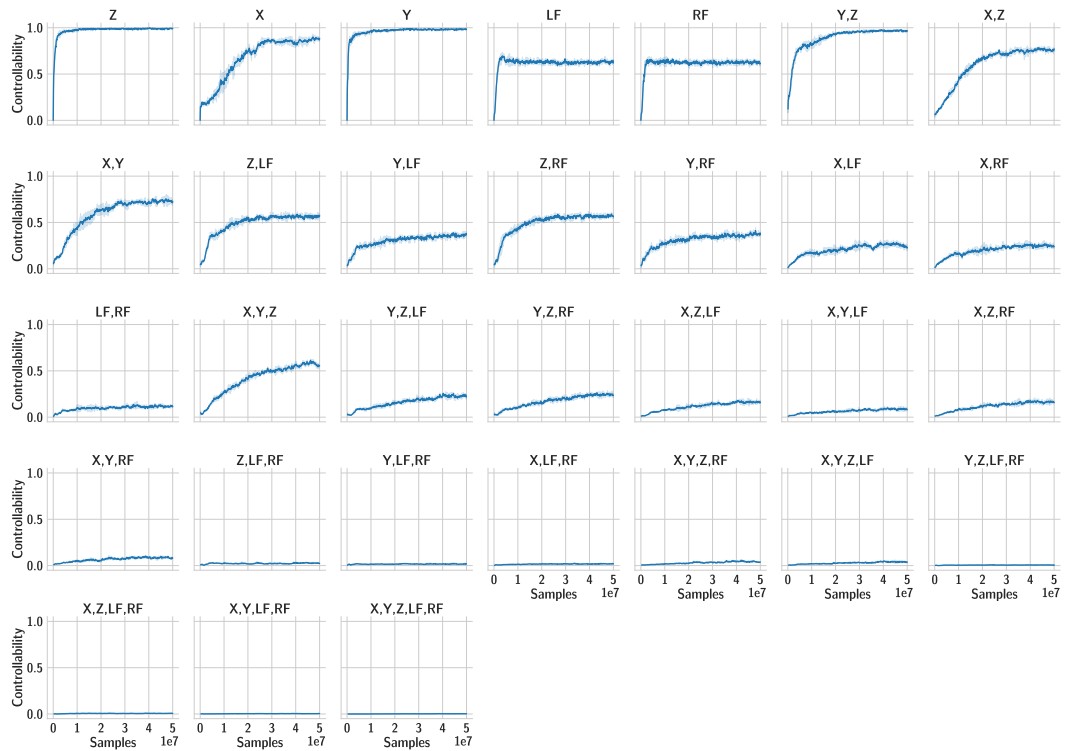

Figure 9: Pre-training performance over the different feature sets considered (Walker robot). Controllability (Y axis) is estimated with a dedicated model. Mean and standard deviation over 3 runs.

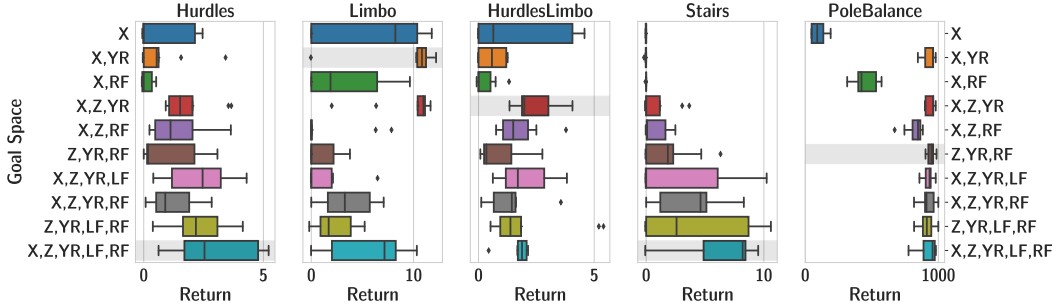

Figure 10: Returns achieved after 50M samples on the benchmark tasks with the Humanoid robot with fixed low-level policy goal spaces (individual runs and quartiles). Each row corresponds to a set of features for the respective goal space. All goal spaces also include the Z rotation of the torso as a feature.

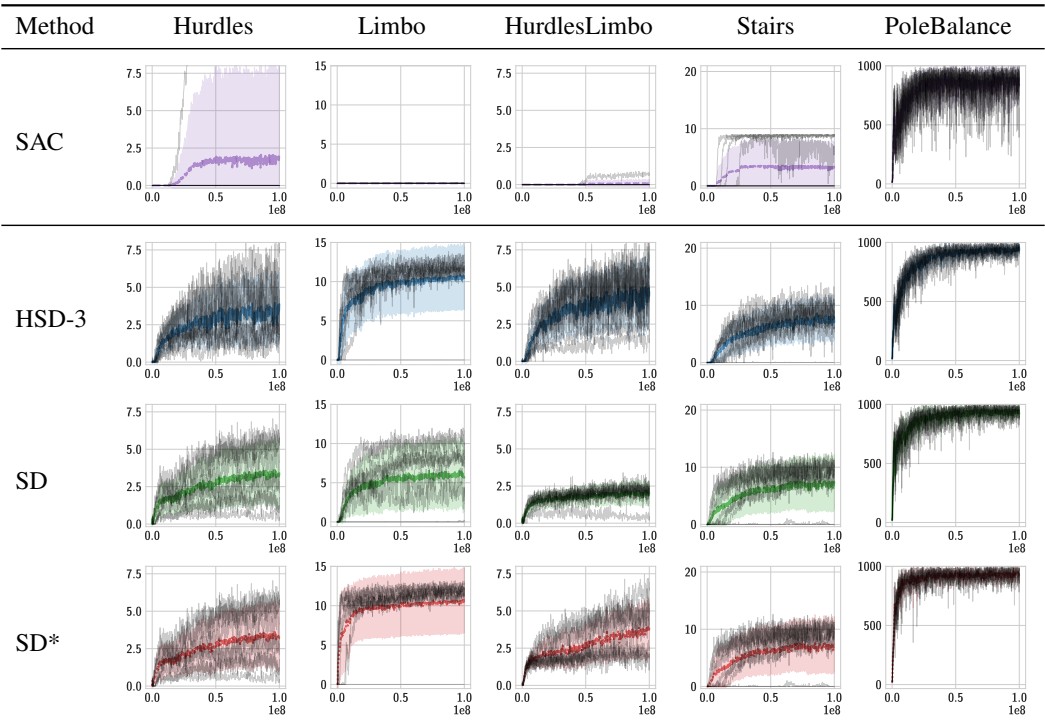

Table 7: Full learning curves for results reported in Figure 6 (Humanoid). We show mean returns achieved (Y axis) after interacting with an environment for a given number of steps (X axis). Shaded areas mark standard deviations computed over 9 seeds for each run.

# G Implicit Priors in HIRO

To highlight the prevalence of implicit priors towards navigation tasks in existing methods, we perform an ablation study on HIRO, a prominent end-to-end HRL method [33, 34]. HIRO defines manual subgoal ranges to steer its low-level policy, where ranges for X and Y locations are of higher magnitude compared to joint angles. In Figure 11, we analyze the impact of normalizing the reward on the AntMaze task. We compute the intrinsic reward for the low-level policy [33, Eq.3] as

$$r(s_t, g_t, a_t, s_{t+1}) = - \left\| \frac{s_t - g_t + s_{t+1}}{R} \right\|_2,$$

with $R = [10, 10, 0.5, 1, 1, 1, 1, 0.5, 0.3, 0.5, 0.3, 0.5, 0.3, 0.5, 0.3]$ corresponding to the high-level policy's action scaling [33, C.1]. The normalization causes all dimensions of the goal space to contribute equally to the low-level policy's learning signal, and effectively removes any implicit prior towards center-of-mass translation. As a result, among the three goals used for evaluation, only the nearby one at position (16,0) can be successfully reached without reward normalization.

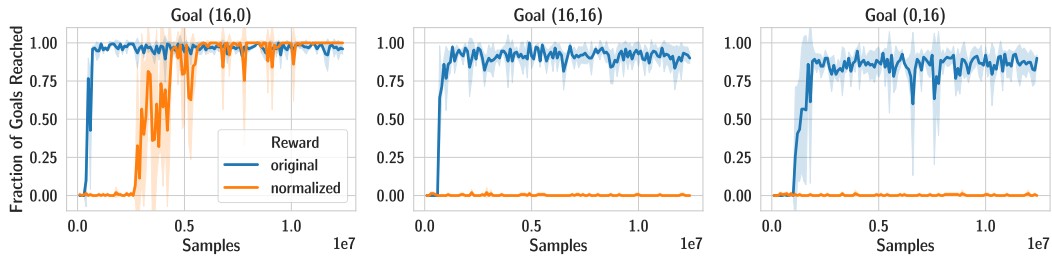

Figure 11: Fraction of goals reached in the AntMaze task with HIRO during evaluations (mean and standard deviation over 3 seeds). With normalized rewards, locomotion on the X-Y plane is no longer subject to higher rewards, and only the nearby goal at position (16,0) can be reached.

Nachum et al. [34] propose to automatically learn a state representation used as the goal space for high-level actions. We investigate the dependence of inductive bias towards X-Y translation by modifying their setup as follows. First, we limit the input to the state representation function $f$ [34, Sec. 2] to the same 15 features that serve as a goal space in [33]. We then normalize the input of $f$ as above via division by $R$, and modify the high-level action scaling accordingly (actions in $[-1, 1]^2$, and a Gaussian with standard derivation $0.5$ for exploration). As a result, the agent is unable to reach any of the tested goals (Figure 12). The limitation of input features alone does not affect performance (blue curve).

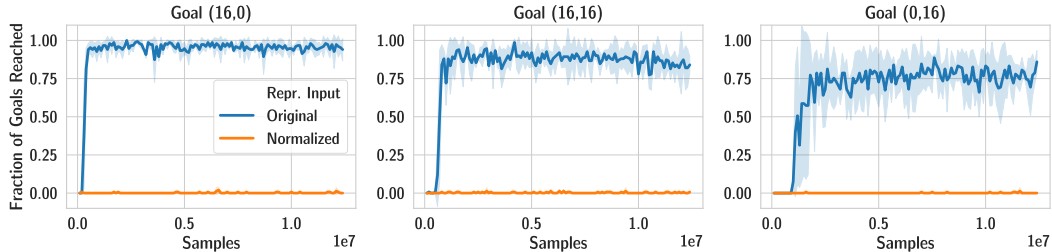

Figure 12: Fraction of goals reached in the AntMaze task with HIRO and representation learning during evaluations (mean and standard deviation over 3 seeds). With normalized state representation inputs, no learning progress is made.