# OpenReview forum: "Hierarchical Skills for Efficient Exploration"
_NeurIPS.cc/2021/Conference — NeurIPS 2021 Poster_

### Official Review · Reviewer_opgb · 2021-06-28

**Rating:** 6
**Confidence:** 4

**Summary:**

The authors propose a 3-level hierarchical method, one that operates on a feature space, another that operates on a feature-conditioned goal space, and finally a low level policy that outputs actions in the environment conditioned on everything above and the current state. They pretrain the policies in a pretraining environment before applying it to the task at hand. The authors also introduce a new suite of environments to test hierarchical RL. The issues brought up by the authors about current HRL methods are enlightening and the method is a novel contribution. Results are fine, with subtle/no improvements over the baselines on some tasks, and substantive improvements on others. I believe the paper has some issues with baseline comparisons and tasks, which I hope will be addressed in the rebuttal. As such, I am currently learning towards not accepting the paper.

**Limitations And Societal Impact:**

Yes.

**Main Review:**

## Paper Strengths
**Paper Framing/Originality**
Pointing out the issues with existing HRL algorithms’ focus on learning skills that are relevant mainly for controlling center of mass is an interesting contribution. Furthermore, the algorithm that results is novel, while also incorporating ideas from modern papers such as the DYNE step-conditioned critic. The inclusion (and supposed release) of the benchmark environments are a benefit to the community as some of them seem like solid tasks to test hierarchical algorithms on.

**Results Significance**
I think the paper does a good job of demonstrating the importance of goal-space separation with results that demonstrate significant improvements over baselines on some of the tasks. Furthermore, it’s rare to see evaluations over 3 seeds, while the authors evaluate on 9 seeds in total. The inductive bias experiment on HIRO in the appendix (sec G) was very enlightening.
The experiment shown in Figure 3 is also very comprehensive and informative.

**Clarity**
Overall, the writing is generally clear and mathematical derivations are pretty well explained.

## Paper Weaknesses

**Sparse-reward tasks**
I think more analysis on similar environments with different robots should be given. Tmake general claims about this method, there should be examples of many robots, not just two. Furthermore, the demonstrating an experiment like Figure 3 for another robot (perhaps the Humanoid that’s already included) would be helpful.

**Experiments**
There should be comparisons with some works which learn skills without explicit assumptions of skill/task types and do not require pre-training skills, e.g. HIDIO (Hierarchical Reinforcement Learning By Discovering Intrinsic Options, Zhang et al. 2021), HiPPO (Sub-policy Adaptation for Hierarchical Reinforcement Learning, Li et al. 2020). These comparisons would further demonstrate the advantages of having the initial pretraining environments (which the comparison to SAC-HIRO already contributes to) and explicit goal space learning when compared with more modern HRL methods, both of which have demonstrated improvements over HIRO in their experiments. In a sense, HIRO is the only true SOTA HRL baseline in this paper so this would need to be addressed.

There’s little explicit analysis that studies how exploration is explicitly affected by these skills, despite it being stressed in the introduction. This can be partially remedied by adding more analysis in Section 5.1 regarding exploration specific to each environment (I think Sec 5.1 is also just generally lacking more analysis).


**Clarity**
I think that an example of the feature set F should be given earlier in the paper, it’s confusing to learn bits and pieces about feature sets F throughout section 3 before getting an example (perhaps at L108). This also makes the distance-based reward (L135) clearer when introduced.

You should bold or highlight the best performances for each column in Table 1. That would improve clarity here greatly.


**Minor Issues**
Some grammatical hiccups throughout the 3rd paragraph of the introduction, making it a little harder to read.


Figure 4: “averaged over 0.5M environment steps” this should be 5M steps

L271: “for SG” -> “for SD”? Or for subgoals?

Appendix L597, equation:  F -> |F|

## Questions for the authors

How do you anticipate extending this to visual environments?

Why are you normalizing the entropy of $\pi^g$ by $|F|$? Isn’t the size of $F$ fixed?

It seems that many of the fixed skill experiments achieve performance nearly on par with full HSD-3. Why is this the case when it seems in Figure 5 many skills are needed?


--UPDATE--: raised score from 5 to 6 after response


**Time Spent Reviewing:**

6

---

> ### Author Response · Authors · 2021-08-09
> **Response to Reviewer opgb**
>
> We thank the reviewer for their valuable feedback and insightful comments. We will address the potential weaknesses that were pointed out individually:
>
> **Sparse-reward tasks:** In general, we view our benchmark task as a first installment and to be potentially extended with further tasks and robots in the future as we agree that methods should be benchmarked in as many scenarios as possible. For this work, we limited our focus on bipedal robots of different complexity. The experiment on individual goal spaces (Figure 3) has also been performed for the Humanoid, although we limited the number of candidate goal spaces to those including translation in X direction. The results are provided in Figure 8 in the Supplementary and paint a similar picture (no single best goal space across tasks).
>
> **Experiments:** The extended results in the Supplementary (Table 5) include an additional end-to-end baseline (Switching Ensemble, from http://arxiv.org/abs/1909.10618). While it doesn't learn a high-level policy, it has been shown to be on par with learning a set of discrete low-level options in http://arxiv.org/abs/1909.10618. Regarding HiPPO, the experiments in the original paper (https://openreview.net/forum?id=ByeWogStDS) are performed in environments where PPO works well already, while non-HRL methods seem to fail almost completely in our tasks. We hence did not consider it as a baseline. HIDIO was published only recently (ICLR 2021). In quick preliminary experiments on Hurdles, Stairs, and GoalWall, using the actions as input to the discriminator, HIDIO did not make meaningful progress in 5M steps apart from learning to not fall over. We will do additional runs using the parameter ranges and variants listed in the paper, and add it as a baseline for the camera ready.
>
> **Clarity:** We provide an example of our skills in the introduction, but will also pick it up again in Section 3 for clarity. We will also address the minor issues that the reviewer helpfully pointed out.
>
> **Questions:**
> - *Visual environments:* Image observations don't directly allow for definitions of goal spaces as in our work. We think that investigating whether these goal spaces can be learned would be worthwhile, as a means to both remove the remaining prior of manual goal space in our method, and to tackle more complex state spaces such as images. Another possibility would be unsupervised keypoint extraction, e.g., as in "Unsupervised Learning of Object Keypoints for Perception and Control" (http://arxiv.org/abs/1906.11883), and to construct goal spaces around those keypoints. For the particular environments in our paper, it would also be possible to restrict pre-training tasks to proprioceptive inputs and supply the high-level policy with image observations of downstream tasks, such as in "Hierarchical Visuomotor Control of Humanoids" (https://openreview.net/forum?id=BJfYvo09Y7).
> - *Normalization of entropy:* The size of \mathcal{F} and the total number of goal space features is fixed, but the subsets F \in \mathcal{F} differ in size. For example, for F={xpos}, there is only one continuous action for \pi^g, while for F={xpos,zpos,left_foot} there are 4 actions (left_foot corresponds to two actual features). We normalize the entropy by |F| to not bias the Q-function towards large feature subsets, which would be the case for the standard SAC formulation.
> - *Many skills achieve good results:* All our individual skills are goal-based policies in a continuous goal space and can be used to express a relatively large range of different motions on their own. For example, it's possible to find goal sequences to move forward in X direction by just controlling a single foot (Fig. 3). This is because the skill policy needs to balance the agent to not fall over, and other features, like the torso position, are not constrained to specific values in this case. Finding these goal sequences can however be challenging, which is demonstrated by the mediocre performance of HSD-Bandit. HDS-3 is free to switch between different skills (i.e., goal spaces), and in Figure 5 we demonstrate that semantically meaningful switching sequences can arise. Here, we don't place any constraints on finding a small set of best goal spaces.

---

> > ### Comment · Reviewer_opgb · 2021-08-10
> > **Response to authors (1)**
> >
> > Thanks for clarifying my questions and addressing some of my concerns. Regarding some of your comments, I have a few things to point out.
> >
> > > Regarding HiPPO, the experiments in the original paper (https://openreview.net/forum?id=ByeWogStDS) are performed in environments where PPO works well already, while non-HRL methods seem to fail almost completely in our tasks. We hence did not consider it as a baseline.
> >
> > I don't think that its experiments being performed in environments where PPO already works well implies that it'll fail on your tasks. In HIDIO's experiments, in which I believe they run an unmodified version of HiPPO, HiPPO performs well on some tasks where flat policies fail. In fact, some of the tasks in HiPPO are semantically similar to the ones evaluated in your environment (there's a "Block Hopper" and "Block Half-Cheetah" in HiPPO). However, I think it's OK to just include one modern hierarchical baseline, which HIDIO satisfies. Specifically regarding HIDIO:
> >
> > > In quick preliminary experiments on Hurdles, Stairs, and GoalWall, using the actions as input to the discriminator, HIDIO did not make meaningful progress in 5M steps apart from learning to not fall over. We will do additional runs using the parameter ranges and variants listed in the paper and add it as a baseline for the camera ready.
> >
> > Can you include a preliminary table of results in a future response to this comment? I understand it's hard to setup your environment tasks and perform a full hyperparameter search for any method with an unfamiliar codebase in a short amount of time, so feel free to just put preliminary results, but not having the extra baseline is my primary concern and I would like to see it fully addressed.
> >
> >
> > Furthermore, the authors did not address this concern of mine:
> > > There’s little explicit analysis that studies how exploration is explicitly affected by these skills, despite it being stressed in the introduction. This can be partially remedied by adding more analysis in Section 5.1 regarding exploration specific to each environment (I think Sec 5.1 is also just generally lacking more analysis).
> >
> > Could you address this in a response? I think Section 5.1 should be expanded upon, or given limited space, an additional pointer to the appendix added and extra analysis inserted there. It's very interesting as is.

---

> > > ### Author Response · Authors · 2021-08-15
> > > **Response to further question of Reviewer opgb**
> > >
> > > We thank the Reviewer for pointing out further opportunities to clarify our contributions. We are currently performing further training runs with HIDIO and will follow up with a results table shortly.
> > >
> > > Regarding analysis of exploration behavior, we would first like to stress that our sparse-reward tasks require efficient exploration for success in the first place. The results in Figure 3 can be directly related to the learning curves in Table 5 (Appendix). We admit that full learning curves would provide further insight into how different goal spaces affect learning in each environment, and we will add such curves for the results in Figure 3 in the Appendix. Furthermore, we will attempt to visualize the state space covered over time by counting ``unique'' states via pseudo-counts (using a hash function) for all the algorithms in Table 5 (and HIDIO, if possible).

---

> > > > ### Comment · Reviewer_opgb · 2021-08-25
> > > > **Results table for other HRL**
> > > >
> > > > Thanks for the response.
> > > >
> > > > > We thank the Reviewer for pointing out further opportunities to clarify our contributions. We are currently performing further training runs with HIDIO and will follow up with a results table shortly.
> > > >
> > > >
> > > > > Furthermore, we will attempt to visualize the state space covered over time by counting ``unique'' states via pseudo-counts (using a hash function) for all the algorithms in Table 5 (and HIDIO, if possible).
> > > >
> > > > Do you have a follow-up with these results? I would appreciate if the authors are able to add these results table as we are now in the last week of reviewer discussion and this would be taken into consideration in my evaluation of the paper.

---

> > > > > ### Author Response · Authors · 2021-08-25
> > > > > **Results for HIDIO and Exploration Behavior**
> > > > >
> > > > > We thank the reviewer for following up on the additional experiments.
> > > > >
> > > > > ### HIDIO Baseline
> > > > >
> > > > > So far, we have run HIDIO on 4 out of 7 tasks (we left out GoalWall and Stairs for now since these are more difficult, and HurdlesLimbo). The results for different discriminator features are as follows (mean values over 3 seeds, after 5M environment steps, comparable to the numbers in Table 1 of our paper):
> > > > >
> > > > > ```
> > > > > Discriminator feature  | Hurdles | Limbo | Stairs | PoleBalance
> > > > > ================================================================
> > > > > State                  | -0.12   | -0.09 | -0.02  | 81.69
> > > > > Action                 | -0.98   | -0.90 | -0.99  | 128.25
> > > > > StateAction            | -0.09   | -0.01 | -0.04  | 112.01
> > > > > ```
> > > > >
> > > > > Negative returns are obtained if the Walker falls over in the course of an episode. We've used the hyper-parameters for the Pusher/Reacher experiments in the HIDIO paper, and are currently sweeping the parameter ranges given in D.1.3 on the Hurdles task. We will follow up with the results from the sweep within the next two days.
> > > > >
> > > > > ### Analyzing Exploration
> > > > >
> > > > > We ran an experiment with both (non-hierarchical) SAC and HSD-3 on the GoalWall environmet (one of the most challenging tasks of the benchmark suite), and counted the number of unique states using the SimHash method from http://arxiv.org/abs/1611.04717. After 5M steps, HSD-3 visited roughly twice as many unique states compared to SAC (400k vs. 200k, https://imgur.com/a/qqHVW6P). We will generate these curves for the remaining methods and tasks for the updated version of the paper.

---

> > > > > > ### Comment · Reviewer_opgb · 2021-08-27
> > > > > > **Thanks for the results**
> > > > > >
> > > > > > Thanks for your response. Please do update this response whenever you get more results, but I am now satisfied that you have addressed my main complaint and will be raising my score in response, assuming these results will be finalized and added into the paper.

---

> > > > > > > ### Author Response · Authors · 2021-08-30
> > > > > > > **Additional HIDIO results**
> > > > > > >
> > > > > > > We thank the reviewer again for this discussion. Sweeping over the hyper-parameters provided in the HIDIO paper (discrimator input: {state, state-action, action, state-difference}, latent vector dimension {8, 12}, rollout length {25, 50, 100}, replay buffer length per parallel actor {50000, 200000}, 3 different runs (seeds)) on the Hurdles task resulted in a few combinations which achieved small positive returns during evaluations. In the table below, we list the maximum evaluation return of those combinations over the course of training (8 million steps). For comparison, both SD and HSD-3 achieve returns above 12 consistently.
> > > > > > >
> > > > > > > | discr input      |   latent dim |   rollout length |   replay buffer length |   run |   max return |
> > > > > > > |:-----------------|-------------:|-----------------:|-----------------------:|------:|-------------:|
> > > > > > > | state_difference |           12 |              100 |                 200000 |     0 |         0.18 |
> > > > > > > | state_difference |           12 |              100 |                 200000 |     1 |         0.02 |
> > > > > > > | state_difference |           12 |              100 |                 200000 |     2 |         0.02 |
> > > > > > > | state_difference |           12 |               50 |                 200000 |     1 |         0.68 |
> > > > > > > | state_difference |           12 |               50 |                 200000 |     2 |         0.34 |
> > > > > > > | state_difference |            8 |               50 |                  50000 |     1 |         0.38 |

---

### Official Review · Reviewer_tdmg · 2021-07-07

**Rating:** 7
**Confidence:** 3

**Summary:**

This paper presents a hierarchical framework for training locomotion policies for tasks with sparse reward. Low level policies are trained to achieve a selection of subgoals and high level policies are trained by exploring in the goal spaces. Challenging locomotion tasks are introduced to show the effectiveness of the proposed methods.

**Ethical Concerns:**

None.

**Limitations And Societal Impact:**

Yes, limitation is discussed.

**Main Review:**

Originality: This paper presents a three level hierarchical policy with a low level policy generated via training on general tasks. This is new to me.

Quality: The paper is technically sound to me.

Clarity: Paper is well written.

Significance: The result is not very impressive since most tasks can be solved via a simple forward progress reward. But it presents an important step towards solving tasks with sparse reward using a hierarchical policy.

Pro:

1. The training of low level policies based on goal spaces specified by position of torso and foot.

2. A three level hierarchical policy and an extension of SAC to accommodate the resulting action distribution.

3. Challenging sparse reward tasks for locomotion.

Con:

1.  The computation required for training even the low level skills are significant. It will be nice to provide more details on the low level training. The low level skills do not seem challenging to me (at least for the walker) so it seems 3 days on 2 GPUs is a lot.

2. Tasks such that designing a simple dense reward (such as the forward progress reward) is none trivial will really demonstrate the strength of hierarchical framework. e.g., the GoalWall task presented. More in Question 2 below.

Question:

1. It is not clear to me why Gap will be more challenging than other tasks. Maybe the reward design needs to be modified? e.g., a negative reward for touching the gap can cause the robot want to terminate as soon as possible. Some failure mode of the gap task will help illustrate the issues.

2. Most of the tasks can also be solved via a simple forward progress reward (with the exception of GoalWall, which is difficult to solve in the current framework anyway). It will be interesting to see what is the performance gain the proposed approach has over baseline methods under this setup.

3. In Figure 6, SAC makes no progress on Stairs, but that is not the case in Table 5 in the Appendix. The video also shows SAC makes some progress on Stairs.

**Time Spent Reviewing:**

3

---

> ### Author Response · Authors · 2021-08-09
> **Response to Reviewer tdmg**
>
> We thank the reviewer for their valuable feedback and comments. We will address the smaller issues pointed out in the write-up and provide detailed answers to the listed cons and questions below:
>
> **Comutational requirement for pre-training:** For the Walker, we use 10M steps for pre-training a shared policy for all goal spaces considered. With uniform sampling over goal spaces, this leads to an expectation of about 322k environment steps per goal space, or skill. Learning a single skill (e.g., moving towards an X-position) in an isolated manner with SAC required significantly more samples in preliminary experiments, and this only increases for more complex skills. Considering the fact that we learn skill policies for 31 goal spaces simultaneously, we think that the resource requirements for pre-training are reasonable.
> For clarity, we will also attach per-skill learning curves with the average rewards reached (in training) to the Appendix.
>
> **Task design:** We agree that designing these tasks is non-trivial, and we consider the investigation into further tasks (especially for Humanoid robots) as a worthwhile future endeavor. We believe that the current suite represents a good first step with 7 varied environments.
>
> **Performance on Gaps:** This task is indeed very challenging since a positive reward is only observed if the robot reaches the next platform. If it touches the Gap (slightly lower than the floor), it receives the same -1 reward as for falling over.
>
> **Performance gain over non-HRL with forward reward:** So far, we found that in dense-reward settings (Hurdles, Limbo, HurdleLimbo, Gaps, Stairs with a reward corresponding to the translation in X-direction per step), the final performance of our method and other HRL approaches we investigated is lower than for plain SAC. We believe this is mainly a limitation of using fixed low-level skills. In the future, it would be interesting to study how to adapt the low-level skills on downstream tasks, or how to use the HRL policy purely for exploration to get good initial traces; we believe this to be out scope for this submission though, as the focus is on improved exploration in sparse-reward settings.
>
> **SAC on Stairs:** Only one out of 9 seeds managed to make meaningful learning progress on the Stairs task, which is not enough to significantly move the average reported in Figure 4. It can be noticed in Table 1, and we will add a corresponding remark in the discussion of results in 5.2 for clarity.

---

> > ### Comment · Reviewer_tdmg · 2021-08-12
> > **further questions**
> >
> > Thanks for your response. Some questions remain:
> >
> > 1. I think reviewer opgb raises a good point about analysis of how analysis is affected. It will be good if the authors can comment on this point. And I may adjust my score based on the response to that question.
> >
> > 2. In the response to reviewer ECxq, the authors claim "Our work demonstrates skill learning and usage for a Humanoid robot without supervision from motion capture data, which has not been shown previously outside of navigation tasks." That is not true. Please refer to literature in computer animation. For example, ALLSTEPS: Curriculum-driven Learning of Stepping Stone Skills, by Xie et al can do more challenging locomotion tasks presented here without motion capture.
> >
> > 3. While I understand the scope of the paper is to solve sparse reward tasks, but my question remains, most of the tasks presented can be solved via very simple forward progress reward (this is not a complicated inductive bias). And the only task that I couldn't figure out a simple dense reward alternative is not solvable by the proposed approach. I think this limitation should be addressed, e.g, in the appendix.

---

> > > ### Author Response · Authors · 2021-08-15
> > > **Response to further questions of Reviewer tdmg**
> > >
> > > These are all good questions, and we are happy to answer them.
> > >
> > > 1. For the first point raised, we refer to our answer to Reviewer opgb below ([link](https://openreview.net/forum?id=NbaEmFm2mUW&noteId=X5cBjZt1HE2)).
> > >
> > > 2. We thank you for pointing out the ALLSTEPS reference, which does indeed eschew motion capture data, but in exchange for a dense, carefully designed reward function and task curricula. We will include it in our related work section.
> > >
> > > 3. A forward reward alone is not sufficient for either GoalWall or PoleBalance, and HSD-3 achieves good results in both of them (with the Walker robot). The focus on sparse-reward environments is motivated by the fact that as tasks grow in complexity, coming up with suitably shaped rewards requires an increasing amount of effort. Naturally, deriving good rewards from e.g. demonstrations is another way to tackle complex tasks, but this direction is orthogonal to our work.

---

### Official Review · Reviewer_gM3y · 2021-07-16

**Rating:** 6
**Confidence:** 4

**Summary:**

This paper argues that, in the context of pre-trained low level skills, there is a trade-off between generality and learning speed. It makes two contributions: firstly, it proposes a benchmarking suite of sparse reward tasks that require different motor skills to study this trade off. Secondly, the paper proposes a hierarchical skill learning algorithm that attempts to trade off generality and learning speed directly. The proposed algorithm outperforms reasonable baselines.

**Limitations And Societal Impact:**

The experimental results with the humanoid are limited in terms of the goal spaces and the full goal space baseline appears to work just as well if not better. This suggests that substantially more work is needed to scale up to more complex embodiments. This is acknowledged by the authors.

**Main Review:**

This paper argues that, in the context of pre-trained low level skills, there is a trade-off between generality and learning speed. It makes two contributions: firstly, it proposes a benchmarking suite of sparse reward tasks that require different motor skills to study this trade off. Secondly, the paper proposes a hierarchical skill learning algorithm that attempts to trade off generality and learning speed directly. The proposed algorithm outperforms reasonable baselines.

The proposed benchmark environments are a number of obstacle courses (e.g. stairs, gaps, hurdles) as well as a pole balancing task. The tasks are set up for the 2d walker as well as a 3d humanoid.

The proposed algorithm pre-trains skills first defining a set of features. Then the agent is trained to control subsets of the full feature set. The subsets and target features are randomly sampled during pre-training. Selecting the feature set induces and the target features induce an inductive bias. I would have like to see more discussion of how the features and target feature distributions are selected. As an aside, I think for more complex embodiments, I think using demonstrations or mocap data to define the targets would be an interesting approach. To reuse the learned skills, the skills are frozen and a hierarchical policy is learned that first selects the subset of features to control and then specifies a target.

I think the related works section is reasonably comprehensive but should cite

    Heess et al.  “Learning and Transfer of Modulated Locomotor Controllers.” arXiv Preprint arXiv:1610. 05182.

as an early example of low level controllers in deep RL. Another relevant paper is

    Hasenclever et al. “CoMic: Complementary Task Learning & Mimicry for Reusable Skills.” ICML 2020. http://proceedings.mlr.press/v119/hasenclever20a.html.

that studies (among other things) the same trade-off between generality and learning speed in the context of skills acquired from mocap data.

In a first experiment, the performance with different fixed goal spaces is compared. The results indicate that no single goal space works best on all tasks: there is indeed a trade-off between speed and generality. However it's worth noting that the full goal space works pretty well across tasks. In the other experiments, the proposed methods performs favourably relative to baselines.

Overall, I think this is a nice paper and I am leaning towards acceptance.

Minor comments:
- page 4, section 3.1 last paragraph: I'm a little confused by this. If you don't reset why call the experience segments episode? Isn't it just one long episode at that point?
- Figure 1: I found the $s^+$ notation confusing. Why not $s^t$
- I would quite like to see some videos of the resulting skills and policies. This is something that other papers in this space provide
- Table 1: How many seeds and episodes do your experiments correspond to?
- Figure 6: typo "are now show"


**Time Spent Reviewing:**

3

---

> ### Author Response · Authors · 2021-08-09
> **Response to Reviewer gM3y**
>
> We thank the reviewer for their thoughtful feedback and comments. We kindly refer the reviewer to the detailed video presentation in the Supplementary ZIP archive (file hsd3.mp4) and will add a corresponding reference in the main body. We show rollouts of the pre-training stage and downstream tasks for both robots, and compare the behavior of HSD-3 to several baselines.
>
> Walker experiments were run with 9 seeds each (for methods with pre-trained skills we use 3 pre-training and 3 high-level training seeds). A lower bound of the number of episodes can be derived from the time limit: for the Walker, 5M steps correspond to at least 5000 episodes (time limit 1000) for all tasks except GoalWall (at least 20000 episodes, time limit 250). All environments implement early termination if the robot falls over, so in practice the total number of episodes will be higher.
>
> We will add a motivation for the selection of our goal space features, and will also reference Heess et al., 2016 (skill learning with a navigation prior) and Hasenclever et al., 2020 (highlighting importance of in-domain motion capture data (as a prior), and proposing joint skill learning and downstream task training as a remedy). We chose the s^+ notation to highlight the fact that it represents additional information introduced by downstream tasks.

---

> > ### Author Response · Authors · 2021-08-30
> > **Did our response address your concerns?**
> >
> > Hello Reviewer gM3y, we would be thankful if you can confirm that we addressed your concerns in our response, and let us know if any issues remain. To summarize, in our response we:
> > - pointed to the video in the supplementary material, which contains comparisons of behaviors of HSD-3 and several baselines
> > - clarified the number of seeds (which is present in the main paper but was missing for Table 1).
> > - will cite Heess et al., 2016 and Hasenclever et al., 2020 in the related work section.

---

### Official Review · Reviewer_ECxq · 2021-07-16

**Rating:** 6
**Confidence:** 3

**Summary:**

This work proposes a benchmark task with bipedal robots instead of locomotion tasks, where the bipedal robots are low dimensional and perform various movements. It also proposes a hierarchical reinforcement learning method with three levels: a policy that specifies a goal space (a set of features to operate on), a policy to specify the goal configuration given that goal space, and a low level policy to reach the desired goal configuration. It learns the low level policies through unsupervised learning, and then optimizes off policy the high level options by optimizing the value function.

**Limitations And Societal Impact:**

This work has a limited discussion of limitations, particularly how closely the method is tied to the task being performed. It has no discussion of societal impact, though robotics has a large variety of effects on modern society so a discussion could have occurred.

**Main Review:**

Abstract: It is not clear throughout the abstract what the "inductive bias" is. It is not clear how the "the potential to facilitate exploration" is an inductive bias. since this is the primary distinction on which this work centers, this makes the abstract confusing.

Introduction:
24 While it is true that much of the work in exploration with hierarchical reinforcement learning deals in navigation or navigation-like domains where the center of mass of the agent is particularly useful, there also exists a large body of work which deals with exploration related to object manipulation where this assumption is not the case. Thus, the generalization expressed here is misleading, since it highlights that "the work that benefits from center of mass knowledge has an implicit bias towards center of mass knowledge."

30 While this is true of policies, it may not be the case that low level skills are more widely applicable, since individual skills, low level or high level might be limited to a a very small set of behavior. It would probably be more appropriate to refer to this in the context of policies.

Related work:
77 While re-usability across different high level actions is a significant benefit, this underscores the idea that one of the large benefits of Hierarchical reinforcement learning is for exploration. Even if a primitive is only used for one task, it might be that the learning procedure is able to exhibit gains in performance or sample efficiency even without generalizability. While it is likely that navigation tasks as a test domain does introduce implicit biases, it is not clear from the description in this section why this is the case.

Hierarchical skill learning:
The "introduction" section for this component lacks a heading, and it is not clear exactly what it is expressing. Is this part a description of the low level policies? A description of the state space or of general terms? At this point it remains unclear what exactly the high level policy comprises, and why there are task specific features and additional objects separated out to be accessed only by the high level policy. While these components gain some clarity later, the ordering makes this confusing.

107 It is not clear up to this point if the argument is being made for more specific, or less specific policies. However, while such a low-level policy as reaching a goal configuration after short time is not navigation related, it is certainly very specific to robotic motion.

3.1 While it is reasonable to train low level skills over different sets of features, perhaps the most important property related to these would be how they are chosen. It is not clear if these features are selected as random samples or specified. However, random sampling seems like a suspect way, as if the number of features exceeds even a small amount the number of possible sets explodes exponentially.

Training the low level policies only by unsupervised pre-training also seems like it could introduce issues. In particular, while the low level space might afford many bad ways of controlling the agent, there are probably only a limited number of useful ways to control the agent. There should be a tradeoff between using the higher level policies to specify what is learned at the lower levels, and simply exploring with the lower level policies, but at present the former appears to be completely ignored.

3.2 Equation numbers would be much appreciated in this section, especially since there are clear changes being made to the typical bellman equation/value function, not the least of which is in the notation of taking the feature mean.
166 While matching sign seems like it should have an effect on optimization, by negating the log of the features, this seems to change the meaning of the equation.

170 It is not entirely clear how the equation in line 164 arrives at the one in 170. In particular, it seems that the loss is the negative expected value, but then the log|F| component has disappeared.

173 A more in-depth description of how the \alpha and \beta loss terms are defined is necessary. At this point, it is simply provided as a given without clear explanation, especially since the intuitive meaning of H^f, H^g is not made clear.

Benchmark Environments:
While these environments are provided as one of the clear contributions of this work, they are described fleetingly. It would be useful to note why properties like center of mass or other normal navigation "implicit biases" do not apply to these cases.

Experimental results:
211 With only 5 features, these proposed domains do not actually differ too significantly from other mujoco locomotion tasks that are more commonly given. As highlighted before, the number of features also seems necessary for this method to work since the subset sampling would explode exponentially otherwise.

227 It is still unclear what emphasis is being made about "no single skill" In particular, it should be expected in any skill learning framework that one skill does not dominate, otherwise there would be no point in using the framework at all.

Figure 4: it is hard to parse the results, in particular those of SD*. Is this meant to outperform the proposed method?

5.2 The proposed baselines, while interesting, do not always capture a fair comparison. In particular they are all single level skill learning methods except for DIAYN, but in this case DIAYN-C does not appear to be ideal for this case.

Figure 6: It is not entirely clear why SAC would completely fail for the given tasks. It would be useful to see a comparison against SAC where it is able to learn at least some useful behavior, or find another baseline that does give valid results. There should exist algorithms which function on humanoid walker.

Overall, this work proposes an interesting way of selecting features to control over and an effective 3 level hierarchy which has encouraging results. While the idea of selecting features has been proposed, it has not been shown in a multi-level hierarchy. However, the writing is sufficiently difficult to parse such that it is difficult to determine how exactly this method is novel from existing work, except that it encodes more specific information for the tasks. It is also difficult to determine from the writing exactly what features of the proposed method contribute to the success. Furthermore, the experiments are limited because they test on a new domain against baselines that do not seem like fair comparison on that domain.

Originality: marginal

Quality: marginal

Clarity: poor

Significance: below marginal


**Time Spent Reviewing:**

3

---

> ### Author Response · Authors · 2021-08-09
> **Response to Reviewer ECxq**
>
> We thank the reviewer for the extensive commentary and suggestions. We group the issues that were raised and respond to them individually.
>
> **Inductive bias:** What we call inductive bias is the prior that is used for exploration on downstream tasks, and hence represents a priori knowledge about what comprises useful behavior in a given environment. Settling for more generality will make pre-trained skills applicable to a larger class of environments, but exploration will be more challenging. Hence, a trade-off arises. For example, controlling the center of mass is effective in navigation environments but not helpful for kicking a soccer ball. This would intuitively require control of the robot's extremities, which is again not useful for exploration in navigation tasks. We will try to present this in a clearer manner in the introduction.
>
> **Novelty:** We clearly work out the role of priors in pre-trained skills, and propose a novel three-level architecture to effectively tackle the trade-off that arises when introducing these priors. Our work demonstrates skill learning and usage for a Humanoid robot without supervision from motion capture data, which has not been shown previously outside of navigation tasks.
>
> **Baselines:** The failure of SAC is explained by the sparse-reward nature of our tasks that require agents with effective exploration capabilities. Occasional SAC runs do make progress on the Stairs task, and after a longer training time on Hurdles (Table 5 & 6). In Table 5, we also give results for the Switching Ensemble (from http://arxiv.org/abs/1909.10618), which improves exploration for SAC and finds effective solutions occasionally. We believe that this answers the reviewer's point on providing a simple baseline that works at least sporadically. DIAYN-C embeds a fixed number of discrete skills in a continuous space, and can hence interpolate between them (http://arxiv.org/abs/1807.10299); we regard it as a superior formulation of DIAYN. In contrast to the baselines, our method is novel in its usage of multiple skills, with each one implemented as a goal-based policy. We are not aware of any similar multi-skill algorithm that would be applicable to our scenario. SD* is considered a topline because it requires exhaustive evaluation on a downstream task -- it is the best goal space, selected a posteriori.
>
> **Benchmark Tasks:** While locomotion is an integral part of the majority of tasks, it is not sufficient to perform well across all of them. This is demonstrated in Figure 3: controlling the center of mass roughly corresponds to controlling X,Y and Z features which does indeed work well on 4 out of 7 tasks, but works poorly on the other 3. The Gaps task, for example, clearly requires control of at least one foot. Further, we can't completely follow the connection drawn by the reviewer between the number of goal space features and the similarity to existing MuJoCo locomotion tasks. The tasks are defined irrespective of the goal space features, and the features have been selected to enable a variety of behaviors for Walker robots (position of torso and the two lower appendages).
>
> **Unsupervised pre-training; guiding skill learning with high-level policies:** In this work, we focused on the scenario of first acquiring pre-trained skill policies, and then utilizing them in unseen downstream tasks. In preliminary experiments, running HSD-3 with uninitialized skills (i.e., from scratch) did not work well; perhaps this could be mitigated with additional inductive biases on which skills should be learned at what point in time, or with a CoMic-like setup (http://proceedings.mlr.press/v119/hasenclever20a.html).
>
> **Section 3.2:** The equations are taken from the SAC paper (https://arxiv.org/abs/1812.05905v2) and adapted to our setting. In the interest of brevity, we do not include the full derivation of the SAC losses but refer to the original paper instead. We will motivate the presence of two temperature loss terms in the camera-ready version; the main idea is that the two high-level policies are sufficiently different and benefit from independent entropy regularization.

---

> > ### Author Response · Authors · 2021-08-26
> > **Follow-up to Response to Reviewer ECxq**
> >
> > We'd like to follow up on our response and verify whether we clarified the questions of Reviewer ECxq. We are happy to participate in further discussion if any question persists.

---

> > > ### Author Response · Authors · 2021-08-30
> > > **Did our response address your concerns?**
> > >
> > > Hello Reviewer ECxq, we were hoping that you can confirm whether we addressed your concerns in our response, and let us know if any issues remain. In summary, our response:
> > > - clarifies the notion of inductive bias we use in the paper, and the derivation of the value and policy gradients based on SAC
> > > - explains why we consider the comparison to existing baselines to be fair
> > > - clarifies our contributions in terms of novelty and our proposed benchmark suite
> > >
> > > As discussed in our [response to Reviewer opgb](https://openreview.net/forum?id=NbaEmFm2mUW&noteId=UmJN7xl4T4R), we will also expand the list of baselines considered with end-to-end HRL algorithm (HIDIO) for the camera-ready version and add an experiment concerning exploration behavior.

---

> > > > ### Comment · Reviewer_ECxq · 2021-08-30
> > > > **Follow up on response**
> > > >
> > > > Hello Authors, your responses provided valuable insight and some assurance on the clarity of the paper. The expanded list of baselines also addresses one of my primary concerns, and I'm willing to raise my score one point.

---

### Decision · Program_Chairs · 2021-09-27

**Decision:**

Accept (Poster)

**Comment:**

The paper proposes a new way of hierarchical goal-based learning. There have been multiple examples of hierarchical strategies where higher levels set goals for the lower levels. However, often such goals are set in a quite low dimensional space, either because the state space is low dimensional to begin with or because a subset of dimensions, e.g. corresponding to the COM coordinates, are pre-specified to be the relevant ones. The paper proposes that which dimensions are relevant for the goal are task dependent, and so it should be up to the higher level policy to choose which dimensions are relevant for goal-setting. In this way, the higher level policy can make the goal for a skill more general or specific, allow a better trade-off between these factors.

The reviewers had mixed opinions initially, but additional results from the authors convinced some reviewers to update their scores, resulting in a unanimous accept recommendation. Summarizing their opinions:
- Originality: the proposed approach is interesting & original. The authors also propose an original and interesting new suite of benchmarking tasks.
- Technical quality: Initially, the opinions were mixed on quality, as some reviewers deemed important baselines to be missing. With the provision of additional HRL baseline, the reviewers were satisfied on quality.
- Relevance and significance: The problem is relevant for NeurIPS. The results were not super surprising (e.g., the 'full goal space' baseline also worked pretty well across tasks), however, reviewers pointed out the paper might be an important step towards solving sparse reward task.
- On clarity, the reviews were a bit mixed, from 'difficult to parse' to 'clear and well explained', with specific issues pointed out for possible improvement.

Overall, the paper proposes an original new method and (taken into account the new results), sufficiently evaluates it in the context of relevant baselines. The paper could certainly be improved further, but I think as is it will be an interesting addition to the NeurIPS program.

I had one additional minor comment: In the introduction, it is mentioned that "In the large body of work ... on HRL ... relies explicitly or implicitly on prior knowledge that low-level skills should control the center of mass" (lines 22-25). While I agree that this is a common assumption, I don't think it's true for the all current HRL methods as seems implied here (e.g. the option-critic lets the agent control any dimensions, feudal networks do use a subspace but the subspace is learned), and in particular, the reference given for this statement in line 25 are mostly methods where the state consists only of the COM, thus, it's inevitable that a HRL method would control the COM rather than a particular assumption. The HIRO paper, for example, would be a better example.
HIRO: "Data efficient reinfocement learning", Nachum et al.
Option-critic: "The option-critic architecture", Bacon et al.
Feudal Networks: "Feudal networks for reinforcement learning", Vezhnevets et al.